# The poly-SUMO2/3 protease SENP6 enables assembly of the constitutive centromere-associated network by group deSUMOylation

Frauke Liebelt[1], Nicolette S. Jansen[1], Sumit Kumar [1], Ekaterina Gracheva[1], Laura A. Claessens[1], Matty Verlaan-de Vries[1], Edwin Willemstein[1] & Alfred C.O. Vertegaal [1]

In contrast to our extensive knowledge on ubiquitin polymer signaling, we are severely limited in our understanding of poly-SUMO signaling. We set out to identify substrates conjugated to SUMO polymers, using knockdown of the poly-SUMO2/3 protease SENP6. We identify over 180 SENP6 regulated proteins that represent highly interconnected functional groups of proteins including the constitutive centromere-associated network (CCAN), the CENP-A loading factors Mis18BP1 and Mis18A and DNA damage response factors. Our results indicate a striking protein group de-modification by SENP6. SENP6 deficient cells are severely compromised for proliferation, accumulate in G2/M and frequently form micronuclei. Accumulation of CENP-T, CENP-W and CENP-A to centromeres is impaired in the absence of SENP6. Surprisingly, the increase of SUMO chains does not lead to ubiquitin-dependent proteasomal degradation of the CCAN subunits. Our results indicate that SUMO polymers can act in a proteolysis-independent manner and consequently, have a more diverse signaling function than previously expected.

[1] Cell and Chemical Biology, Leiden University Medical Center, Leiden, The Netherlands. Correspondence and requests for materials should be addressed to A.C.O.V. (email: vertegaal@lumc.nl)

Protein posttranslational modifications (PTMs) play key roles in virtually all cellular processes. PTMs include small chemical modifications like phosphorylation and modifications by small proteins belonging to the ubiquitin family[1–4]. Ubiquitin signal transduction includes extensive polymer formation via all of its seven internal lysines[5] and via head-to-tail linear polymers[6]. Ubiquitin polymers are well known for their classical role in targeting proteins to the proteasome for degradation[7]. All ubiquitin chains formed via internal lysines accumulate upon proteasome inhibition, with the exception of K63 linked chains[8]. Ubiquitin chains are disassembled by proteases in a chain-type dependent manner[9].

Small ubiquitin-like modifiers (SUMOs) regulate proteins via mono-SUMOylation, multi-SUMOylation and poly-SUMOylation[3]. SUMOylation can collectively target groups of proteins that are functionally or physically connected, making single SUMO modification events redundant and potentially explaining the simplicity of the SUMOylation machinery, which comprises a modest set of enzymes in contrast to hundreds of enzymes participating in ubiquitin signaling[10]. SUMOs predominantly signal via monomers under regular cell culture conditions involving dynamic deSUMOylation by SUMO specific proteases[11].

Two of the three conjugated mammalian SUMO family members, SUMO2 and −3, are able to efficiently form SUMO polymers via internal SUMOylation sites in their flexible N-terminal domains in vitro[12] and in cells[13,14]. These chains are stabilized or increased by cellular stress, such as heat shock[15]. SUMO E3 ligases have the ability to catalyze automodification by SUMO polymers[16] and are consequently key substrates for the SUMO-chain targeted ubiquitin ligase (STUbL) RNF4[17]. In Saccharomyces cerevisiae SUMO chain formation is regulated by the covalent SUMO attachment to the single SUMO E2 conjugating enzyme, Ubc9. This activity is counterbalanced by the SUMO specific protease Ulp2 that is able to disassemble the accumulated SUMO chains[18,19]. SUMO chains contribute to synaptonemal complex formation during meiosis in yeast[18,20] and are required to prevent aneuploidy[21].

In mammalian cells two members of the SUMO specific protease (SENP) family, SENP6 and SENP7, are responsible for the depolymerization of SUMO chains[22,23]. These proteases predominantly localize throughout the nucleoplasm and possess conserved sequence insertions within their catalytic domain, which are absent from the catalytic domains of the other SENP family members. These insertions are proposed to be responsible for their poly-SUMO2/3 specificity[24–28]. The importance of a balanced regulation of SUMO chains was demonstrated by studies in mammalian cells where SENP6 depletion, and subsequent accumulation of SUMO2/3 conjugates led to severe mitotic problems and reduction in cell survival[29,30]. The identity of the regulated substrates remains largely unknown.

SUMO chains were identified as substrates for STUbLs[31]. These STUbLs were identified in yeast[32–34] and contain multiple SUMO interaction motifs (SIMs), explaining their preference for poly- and multi-SUMOylated proteins[35]. The initially identified substrate for the mammalian STUbL RNF4 was the promyelocytic leukemia protein PML[35,36]. PML and the PML-RARα oncogene product are targeted for degradation by the proteasome upon ubiquitination by RNF4 in response to arsenic trioxide treatment-induced poly-SUMOylation[35,37]. The centromere protein CENP-I was proposed to be regulated in a similar fashion. SENP6 is necessary to trim down the SUMO chain which would otherwise lead to the RNF4-mediated ubiquitination and proteasomal degradation of CENP-I[30].

In contrast to our extensive knowledge on ubiquitin polymer formation, we are limited in our understanding of SUMO polymers, particularly due to limited knowledge of the identity of the substrates modified by these polymers. We set out to identify these target proteins, capitalizing on our developed SUMO purification technology[38] combined with knockdown of the poly-SUMO2/3 processing protease SENP6. We identify several highly interconnected groups of proteins that are regulated by SENP6, indicating a striking group de-modification and involvement of SENP6 in multiple crucial cellular processes. One of the identified interconnected groups regulated by SENP6 represents most of the subunits of the constitutive centromere-associated network (CCAN), including the previously identified subunit CENP-I. Accumulation of poly-SUMO2/3 on CCAN subunits leads to a reduced abundance of these proteins at the chromatin and the centromere. Surprisingly, we fail to observe an accumulation of SUMOylated or ubiquitinated CCAN proteins upon inhibition of the proteasome and RNF4 knockdown, which contradicts the classical consequence of poly-SUMO2/3 accumulation. We conclude that SUMO polymers can also act in a proteolysis-independent manner and therefore have diverse signaling functions.

## Results

**SENP6 is vital for proliferation and cell cycle progression.** SENP6 and SENP7 are the mammalian SUMO proteases with a preference for poly-SUMO2/3 (Fig. 1a). SENP6 is able to rapidly depolymerize SUMO2 chains in vitro, while cleaving di-SUMO moieties much less efficiently (Fig. 1b). Knockdown of SENP6 caused an increase in high-molecular weight SUMO2/3 conjugates, but knockdown of SENP7 did not, whereas combined knockdown of both SENP6 and SENP7 caused a stronger increase in SUMO2/3 conjugates (Fig. 1c and Supplementary Fig. 1d). Since SENP6 was previously proposed to be essential for mitotic progression and cell survival[29,30,39] we aimed to further investigate its function. Knockdown of SENP6 by two independent shRNAs reduced colony formation to a large extent, demonstrating an important contribution of SENP6 to cell proliferation (Fig. 1d). We investigated cell cycle profiles of SENP6 depleted cells and observed an increase of cells in G2/M phase (Fig. 1e, Supplementary Fig. 2). Furthermore, we observed clear signs of mitotic problems. Knocking down SENP6 strongly induced the formation of micronuclei, which are a hallmark for lagging acentric chromosomes during anaphase due to faulty mitotic processes (Fig. 1f)[40]. In conclusion our results confirm that SENP6 is essential for cell proliferation and cell cycle progression with a prominent role during mitosis.

**Identifying target proteins regulated by SUMO polymers.** To obtain global insight into the signaling by poly-SUMO2/3 and the cellular pathways involved, we set out to identify poly-SUMOylated proteins regulated by SENP6. For this purpose, we combined our SUMO2 purification methodology[38] with knockdown of SENP6 by two independent shRNAs in a label-free quantitative proteomics approach (Fig. 2a). U2OS cells stably expressing His10-tagged SUMO2 were treated with lentivirus encoding either a nontargeting control shRNA or one of the two SENP6-targeting shRNAs. Both shRNAs efficiently depleted SENP6 and caused a major increase in high-molecular weight SUMO2 conjugates as well as free SUMO chains (Fig. 2b). SUMO2 conjugates were purified by means of a His10-pulldown and identified by mass spectrometry and quantified using MaxQuant and Perseus software (Supplementary Data 1)[41,42]. Overall, we identified 180 SUMO target proteins enriched at least twofold upon knockdown of SENP6 by each shRNA separately (Supplementary Data 2). Intriguingly, the identified SUMOylation targets regulated by SENP6 included ten

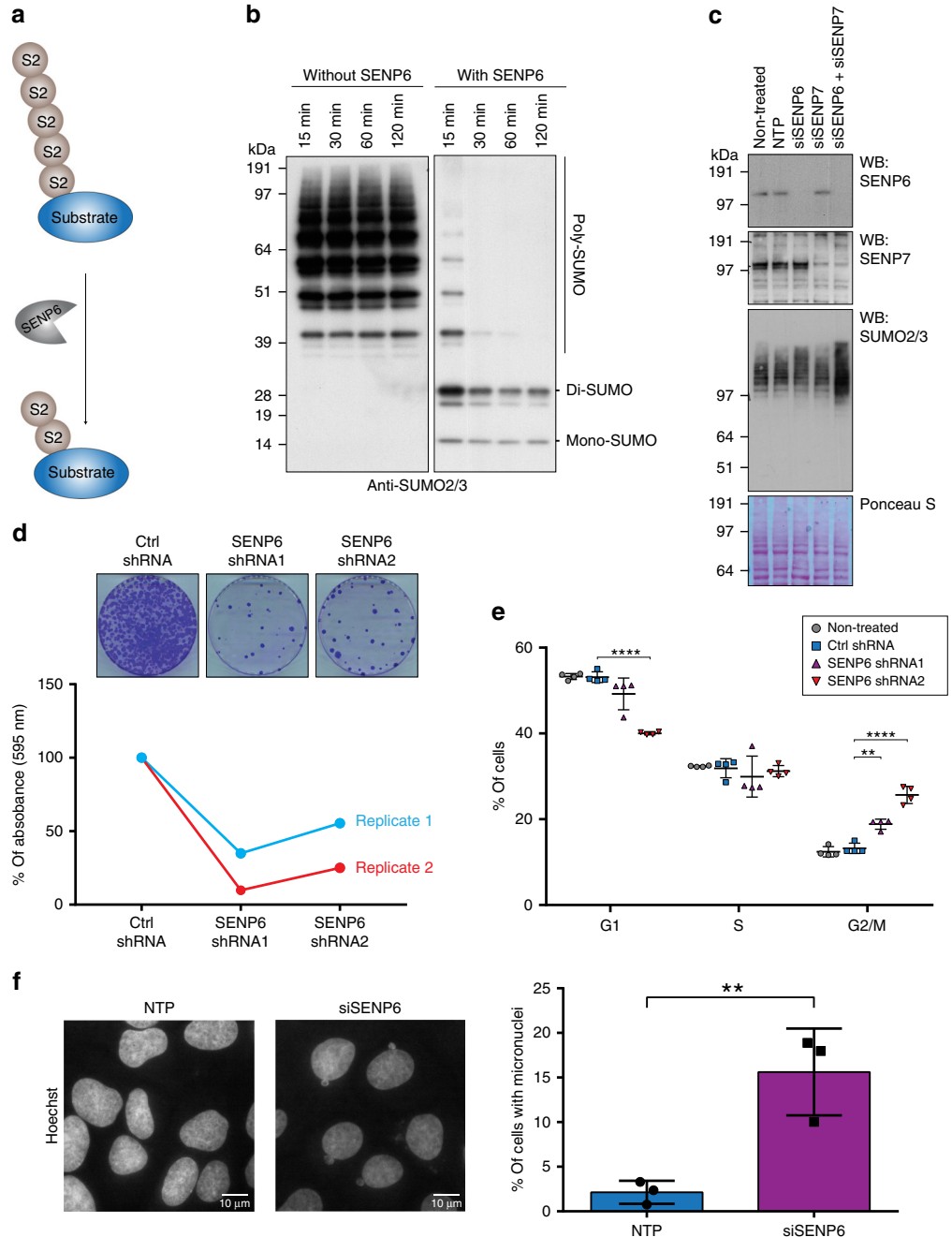

**Fig. 1** SENP6 is important for cell proliferation and cell cycle progression. **a** SENP6 cleaves poly-SUMO2/3 from target substrates. **b** Recombinant poly-SUMO2/3 was treated in vitro with recombinant SENP6 for the indicated time and immunoblotting was performed using a SUMO2/3 specific antibody. **c** U2OS cells were left untreated or transfected with either a pool of four siRNAs against SENP6 (siSENP6), SENP7 (siSENP7), a combination of both or a pool of four nontargeting siRNAs (NTP). Cell lysates were analysed 2 days post transfection by immunoblotting using antibodies against SENP6, SENP7, and SUMO2/3. **d** U2OS cells were infected with lentiviruses encoding shRNAs against SENP6 or a nontargeting control (ctrl) shRNA. Colony formation was determined by crystal violet staining. Line graphs represent the absorbance of solubilized crystal violet of two independent biological replicates ($n = 2$ independent experiments). **e** Scatter plot showing the percentages of HeLa cells in each cell-cycle phase (G1, S, and G2/M) of four biological replicates ($n = 4$ independent experiments). HeLa cells were treated with lentiviruses as in panel **d**. Cells were fixed and prepared for flow cytometry analysis 4 days post infection. Gray circles represent nontreated cells, blue squares represent control (ctrl) shRNA, purple triangles represent SENP6 shRNA1, red triangles = SENP6 shRNA2. Error bars represent standard deviation and p-values are derived from two-sided two samples t-tests and FDR corrected. **$p < 0.01$, ***$p < 0.0001$. **f** U2OS cells were treated either with a pool of four siRNAs against SENP6 (siSENP6, purple bar) or NTP control (blue bar). Cells were fixed for microscopy 2 days post transfection and nuclei were stained with Hoechst. Ten pictures per condition of three biological replicates were taken. Total amounts of interphase nuclei and amounts of nuclei associated with one or more micronuclei were counted. Representative micrographs are shown. Scale bars = 10 μm. The bar graph shows the average percentage of cells that were associated with micronuclei over three biological replicates. Error bars represent standard deviations and the p-value is derived from a two-sided two-sample t-test with $n = 3$ independent experiments. **$p < 0.01$. Source data are provided as a Source Data file

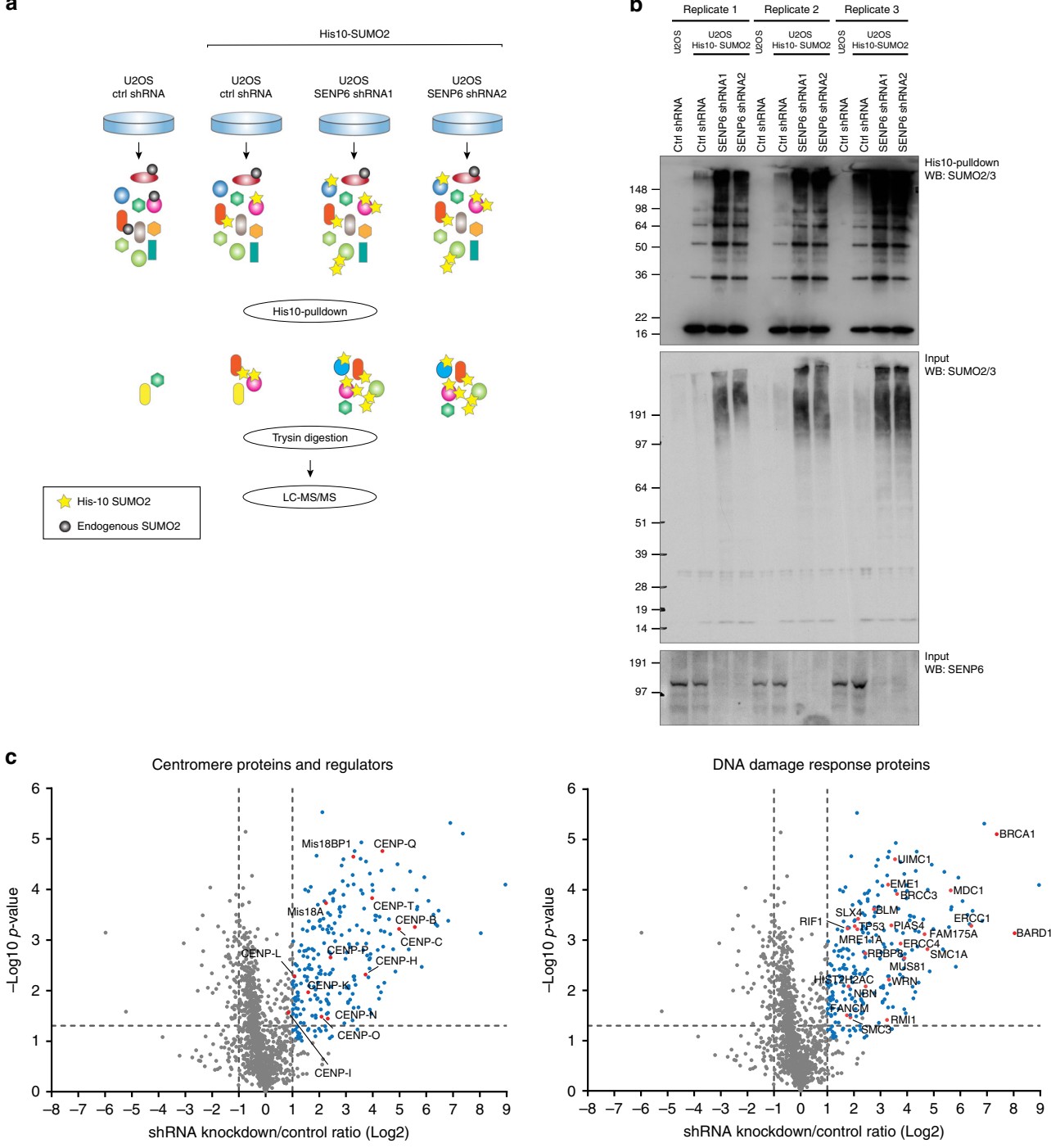

**Fig. 2** Identification of target proteins regulated by SUMO polymers. **a** Experimental set up for the identification of SENP6-regulated proteins. U2OS cells stably expressing His10-SUMO2 were infected with lentiviruses encoding shRNAs targeting SENP6 or a nontargeting control (ctrl) shRNA. Cells were lysed 3 days post infection and SUMOylated proteins were enriched by means of Ni-NTA pulldown. Enriched SUMOylated proteins were trypsin digested and prepared for label-free quantitative mass spectrometry. Peptides were identified by LC–MS/MS. The four experimental conditions of three biological replicates were analysed in two technical repeats per sample, resulting in a total of 24 MS runs. Black circles represent endogenous SUMO, yellow stars represent exogenous His10-SUMO2. **b** Immunoblot analysis of the three biological replicates analyzed by mass spectrometry. An antibody against SUMO2/3 was used to confirm efficient enrichment of SUMO conjugates and an increase of SUMO conjugates upon SENP6 knockdown. An antibody against SENP6 was used to confirm efficient knockdown. **c** Volcano plot showing all identified proteins within the SENP6 knockdown samples compared with the nontargeted control shRNA. Dashed lines indicate a cutoff at twofold change (log2 = 1) and a p-value of 0.05 (−log10 = 1.3), n = 3 independent experiments. SUMOylated proteins represented as blue circles were more abundant after SENP6 knockdown. The left panel shows identified centromere proteins and protein involved in centromere regulation represented by red circles, whereas the right panel shows DNA damage response proteins represented in red circles. Source data are provided as a Source Data file

out of the sixteen subunits of the CCAN, indicating a remarkable group deSUMOylation (Fig. 2c). Other highly regulated SUMO targets included proteins that are associated with the DNA damage response (Fig. 2c).

**SENP6 demonstrates group deSUMOylation activity.** Subsequently, we analyzed the set of SENP6-regulated proteins using bio-informatics. STRING interaction network analysis[43] revealed a large interconnected set of nuclear proteins (Fig. 3a). Highly interconnected subclusters were revealed by the Cytoscape plug-in MCODE[44]. The most interconnected clusters consisted of proteins that are involved in the DNA damage response (Fig. 3b), regulation and assembly of the kinetochore (Fig. 3c), ribosomal RNA metabolism (Fig. 3d), and DNA recombination (Fig. 3e). Previously identified proteins regulated by SUMO chains such as SUMO E3 ligases, the nuclear body component PML, DNA damage response factors BRCA1 and BARD1, and centromeric protein CENP-I were identified in our screen, serving as positive controls[17,29,30].

To identify enriched Gene Ontology (GO) terms within the SENP6-regulated protein population, we made use of the online GO tool (Geneontology.org), focusing on the categories of cellular compartments, molecular functions, and biological processes. The analysis confirmed that the identified proteins were strongly enriched for nuclear proteins functioning in DNA-associated processes like DNA repair, chromosome segregation, and regulation of the cell cycle (Fig. 3f). The highly interconnected networks of proteins that are regulated by SENP6 demonstrate striking group deSUMOylation.

**Poly-SUMO accumulates on CCAN subunits upon SENP6 knockdown.** The overall dynamics of SUMO2/3 modification in the absence of SENP6 were striking, including SUMOylation changes of up to ~900-fold (Supplementary Data 1 and 2). We verified dynamics of SENP6-mediated deSUMOylation using immunoblotting. In agreement with the mass spectrometry data, we confirmed massive buildup of SUMO chains on CENP-B, -C, -H, -I, -K, and -T with SUMO conjugates extending all the way to the top of the protein gels in a manner reminiscent to ubiquitin polymers. Continuous processing of the poly-SUMO2/3 by SENP6 under regular cell culture conditions has prevented the identification of the CCAN subunits as SUMOylation targets up to this moment. Likewise, we confirmed extensive SUMOylation of other mitotic regulators such as KIF18A, which is involved in chromosome congression, Mis18BP1, which is involved in the positioning of the histone H3 variant CENP-A, and KIF23, which plays a role in cytokinesis (Fig. 4a). SUMO chains did not generally build up on all SUMO targets, as shown for DNA topoisomerases IIα and IIβ (Fig. 4b). Consistent with the mass spectrometry results, SUMOylation of CENP-A could not be detected by immunoblotting (Fig. 4c). In vitro SUMOylation of CENP-T with SUMO2 either WT or lysine-less SUMO2 (K0), which is unable to form SUMO polymers, showed that the high-molecular weight signal of SUMOylated CENP-T could be attributed to poly-SUMOylation (Supplementary Fig. 1a). Mass spectrometry analysis of in vitro SUMOylated CENP-T revealed the presence of poly-SUMO2/3 on CENP-T (Supplementary Fig. 1b). Furthermore, we demonstrated that SENP6 is able to directly target poly-SUMOylated CENP-T and depolymerize the accumulated high-molecular weight poly-SUMO2 chain in vitro (Supplementary Fig. 1c).

**Proteasomal degradation-independent function of poly-SUMO.** SUMO chains were previously found to accumulate on CENP-I, PML, and PML-RARα and were proposed to mediate

the recruitment of the STUbL RNF4. Ubiquitination by RNF4 caused their degradation by the proteasome[30,35,37]. We investigated the fate of SUMOylated CCAN family members upon inhibition of the proteasome by immunoblotting. Surprisingly, we noted a reduction in SUMOylation of CENP-K, -T, -I, -C, and -H upon proteasome inhibition instead of an increase as would be expected. However, total SUMO conjugates showed a substantial increase after SENP6 knockdown and proteasome inhibition compared with SENP6 knockdown only (Fig. 5a). In addition, SENP6 knockdown led to a global increase of ubiquitinated proteins within the SUMOylated fraction and SUMOylated proteins within the ubiquitinated fraction that further increased by proteasome inhibition (Supplementary Fig. 3a, b). These observations indicate that a substantial fraction of SENP6-regulated SUMO conjugates are destabilized by ubiquitination and proteasomal degradation, while modified CCAN subunits have a different fate.

To verify that accumulation of SUMO chains on the CCAN subunits may not target them to the proteasome, we investigated the ubiquitinated fractions of some of the CCAN proteins. A small fraction of CENP-K was ubiquitinated under control condition, but neither proteasome inhibition, nor SENP6 knockdown or a combination of both led to a substantial accumulation of ubiquitinated CENP-K (Fig. 5b). The ubiquitination of CENP-T was barely visible by immunoblot analysis after ubiquitin purification, but CENP-T ubiquitination did not seem to be stabilized by SENP6 knockdown while a slight increase was visible upon proteasome inhibition, notably also under control conditions (Fig. 5b).

Our results indicate that there is a small fraction of ubiquitinated CCAN subunits and while we clearly observe an increase in SUMOylation after SENP6 knockdown we fail to observe an increase in ubiquitination of these proteins even after proteasome inhibition. Also, shRNA mediated knockdown of RNF4 did not obviously stabilize SUMOylated CENP-T, -K, and -H (Supplementary Fig. 4a). Taken together, our findings indicate that SUMO chains on the CCAN family members do not act as a degradation signal, pointing towards a nonclassical role of poly-SUMO2/3 signaling. However, shRNA mediated knockdown of RNF4 did stabilize SUMOylated Mis18BP1 (Supplementary Fig. 4a). Moreover, we verified that the knockdown efficiency of RNF4 in our experiments was sufficient to reduce ubiquitination of PML in response to As$_2$O$_3$ treatment (Supplementary Fig. 4b).

**SENP6 knockdown reduces CENP-T and CENP-W at centromeres.** To address the functional consequences of highly increased SUMOylation of the CCAN proteins in the absence of SENP6, we studied their subcellular localization by immunofluorescence (Fig. 6). We focused on CENP-T and CENP-W, which are direct binding partners and together with CENP-S and CENP-X form one of the five subcomplexes of the CCAN[45]. CENP-T is critical for the assembly of other CCAN components except for CENP-C. CENP-T, and -C act in two parallel pathways to recruit the KNL1/Mis12 complex/ Ndc80 complex (KMN) network, which is the microtubule-binding platform of the kinetochore[46,47]. CENP-T, -W, -S, and -X possess a histone fold and together form a nucleosome-like structure that enables DNA binding. Therefore, this complex is an important link between DNA and microtubules[45].

We found that after treatment with a nontargeted siRNA pool (NTP), CENP-T accumulated into bright distinct foci in mitotic and interphase cells as expected, marking the centromere. However, in the absence of SENP6, we noted that centromeric accumulation of CENP-T was reduced in mitotic cells as well as in interphase cells, indicating that deSUMOylation by SENP6 is

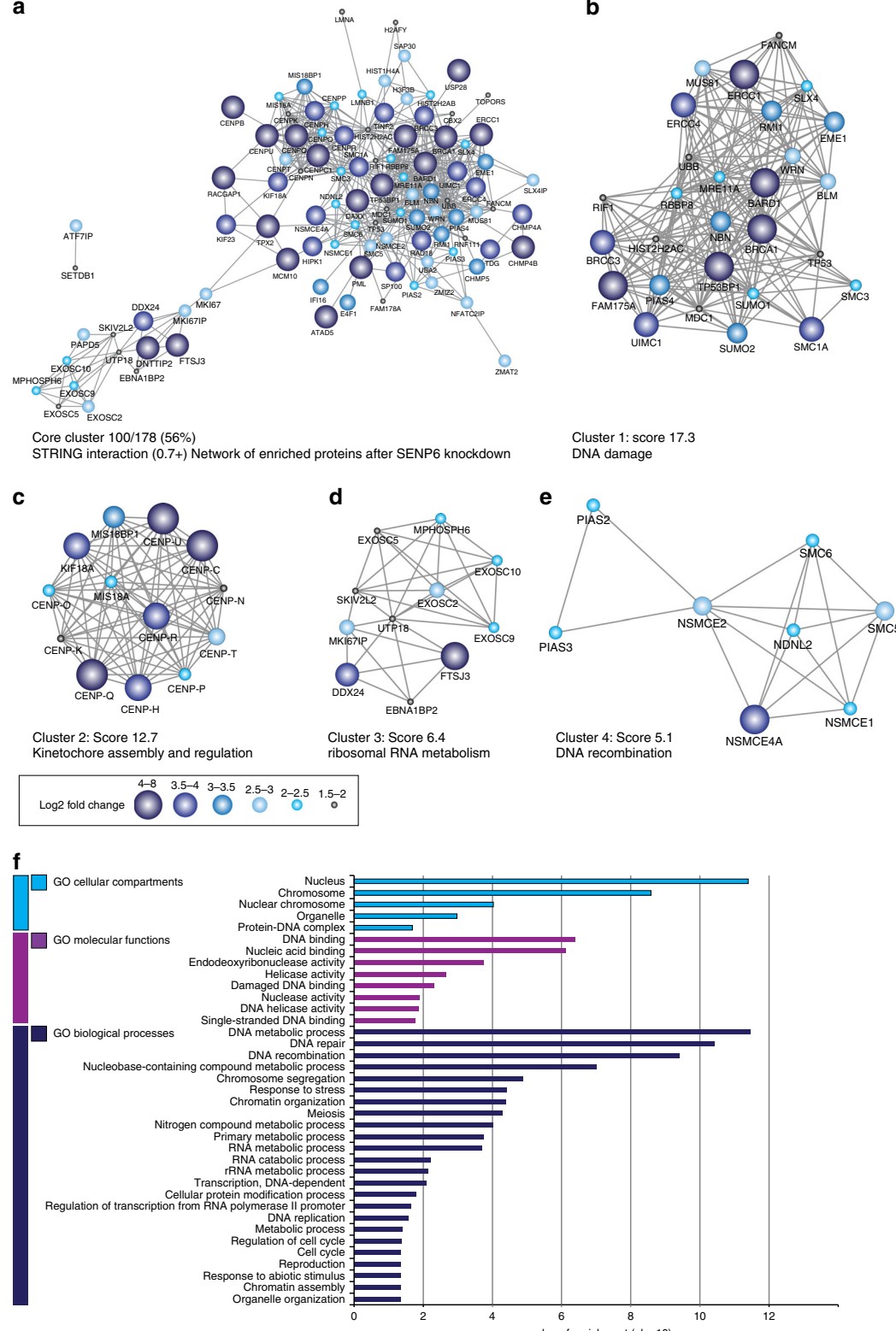

**Fig. 3** SENP6 demonstrates group deSUMOylation activity. **a** STRING network analysis of enriched proteins after SENP6 knockdown, with a STRING interaction confidence of 0.7 or higher. Cytoscape software was used to visualize the interaction network. Color and node size indicate the fold-change differences in abundance after SENP6 knockdown compared with the nontargeting control. **b** MCODE was used to extract the most highly interconnected clusters form the network shown in **a**. Cluster 1 contains multiple proteins involved in DNA damage response. **c** Cluster 2 includes multiple kinetochore and kinetochore-associated proteins. **d** Cluster 3 includes proteins that are involved in ribosomal RNA metabolism. **e** Cluster 4 contains proteins associated with DNA recombination and the SUMOylation pathway. **f** Gene Ontology (GO) enrichment analysis of SENP6-regulated proteins. The bar graph shows the most significantly overrepresented GO terms for biological processes (BP) in dark blue, molecular functions (MF) in purple, and cellular compartments (CC) in light blue compared against the annotated human proteome. Source data are provided as a Source Data file

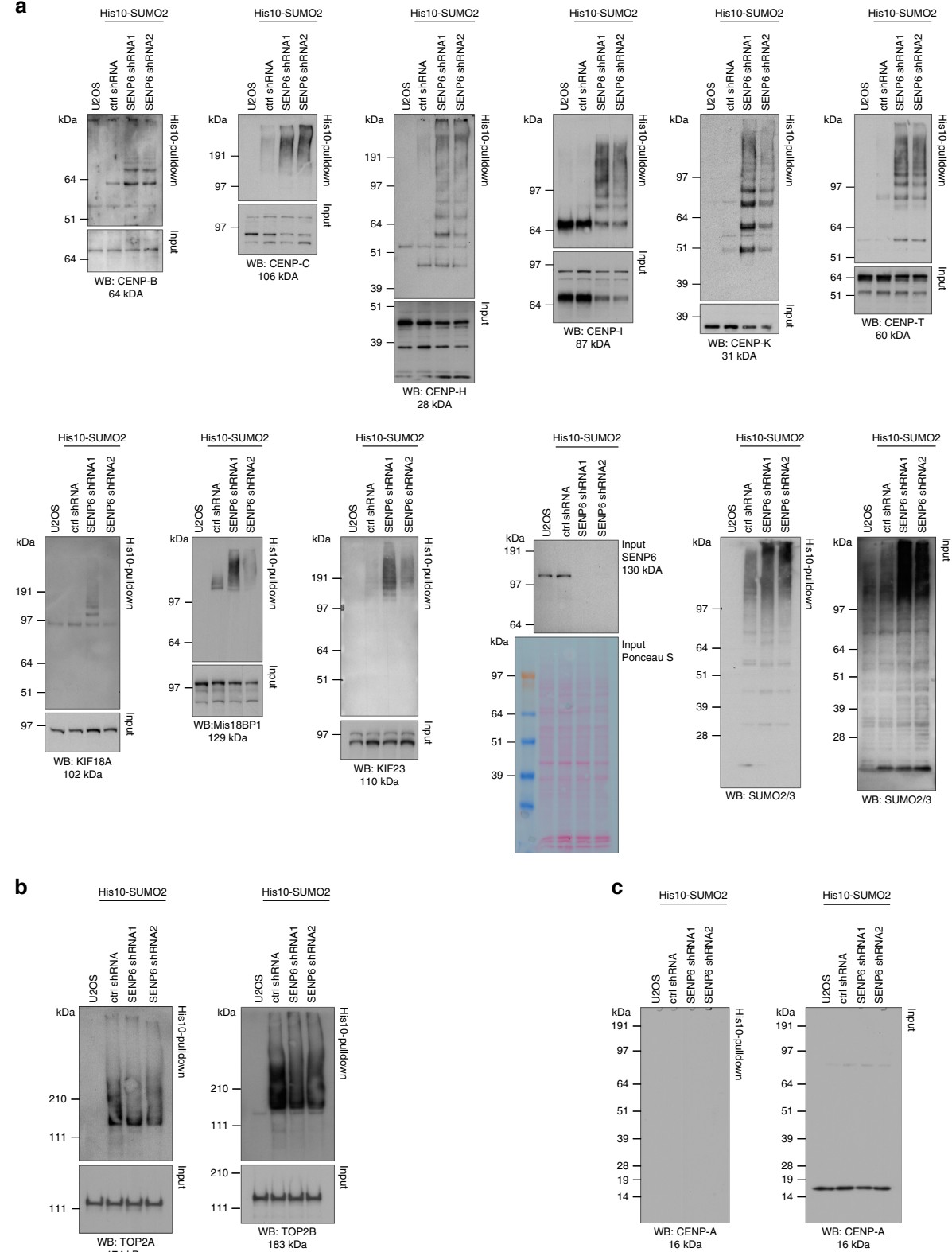

**Fig. 4** Immunoblot validation of proteins identified by mass spectrometry. **a** U2OS cells stably expressing His10-SUMO2 were infected with lentiviruses encoding shRNAs against SENP6 or a nontargeting control shRNA (ctrl shRNA). Cells were lysed 3 days post infection and SUMOylated proteins were enriched by means of Ni-NTA pulldown. Inputs and His10-pulldown elutions were analysed by immunoblotting with the indicated antibodies. SENP6 and SUMO antibodies were used as control for efficient knockdown of SENP6 and increase of SUMO conjugates. **b** Samples as in **a** were analysed by immunoblotting against Topoisomerases IIα (TOP2A) and IIβ (TOP2B) that did not show increased SUMOylation in the mass spectrometry screening. **c** Samples as in **a** and **b** were analysed by immunoblotting against CENP-A. Source data are provided as a Source Data file

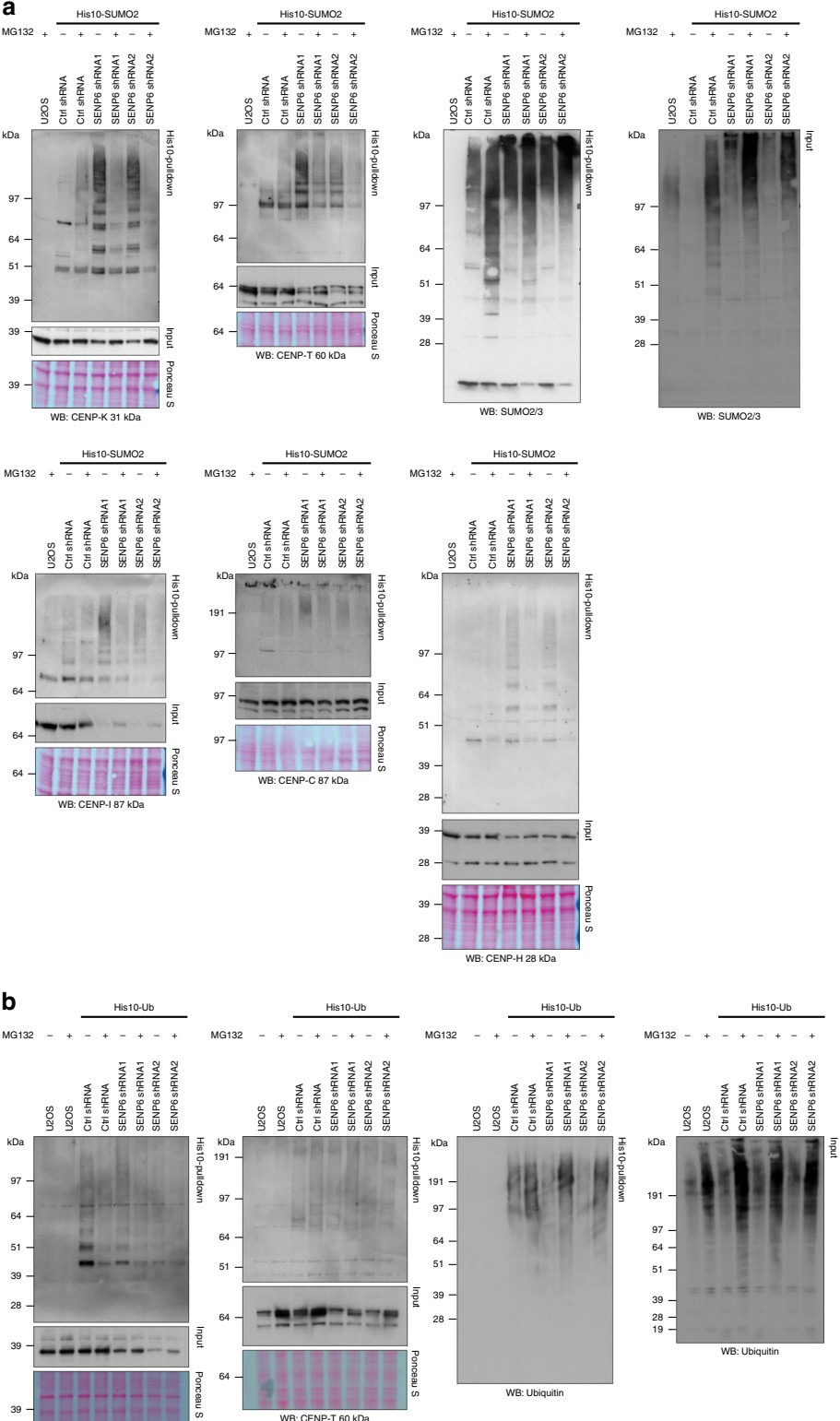

**Fig. 5** Poly-SUMOylation does not lead to destabilization of CCAN proteins. **a** U2OS cells stably expressing His10-SUMO2 were infected with lentiviruses encoding shRNAs against SENP6 or a nontargeting control shRNA (ctrl shRNA) 3 days prior to lysis. Where indicated, cells were treated with 10 μM MG132 for 4 h prior to lysis. Cells were lysed and SUMOylated proteins were enriched by means of Ni-NTA pulldown. Inputs and His10-pulldown elutions were analysed by immunoblotting with the indicated antibodies. Equal loading was verified by Ponceau S staining. **b** U2OS cells stably expressing His10-ubiquitin were infected with lentiviruses encoding shRNAs against SENP6 or a nontargeting control shRNA (ctrl shRNA) 3 days prior to lysis. Where indicated cells were treated with 10 μM MG132 for 4 h prior to lysis. Cells were lysed and ubiquitinated proteins were enriched by means of Ni-NTA pulldown. Inputs and His10-pulldown elutions were analysed by immunoblotting with the indicated antibodies. Equal loading was verified by Ponceau S staining. Source data are provided as a Source Data file

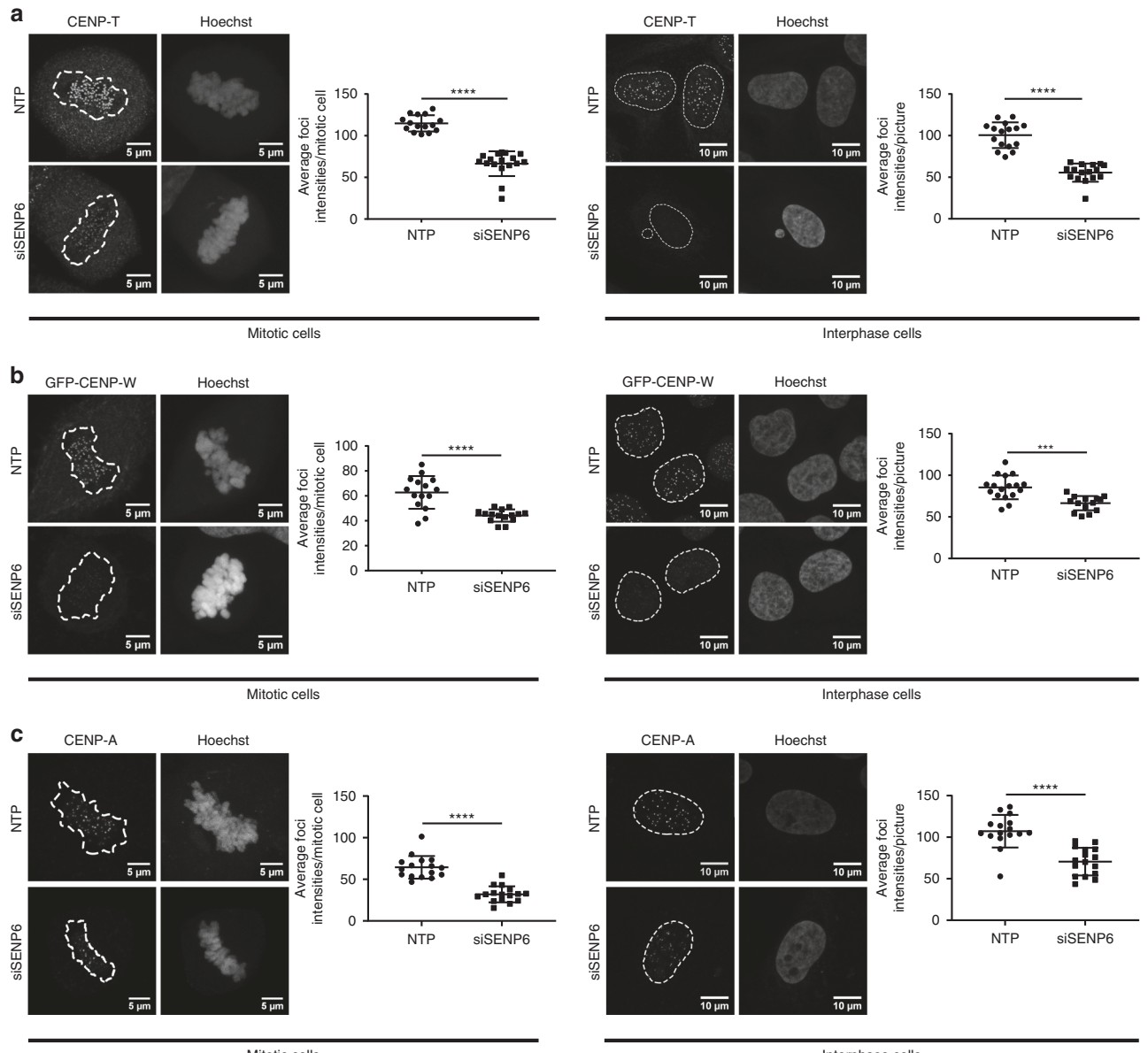

**Fig. 6** Poly-SUMO2/3 prevents accumulation of CCAN proteins at centromeres. **a** U2OS cells were transfected either with a pool of four siRNAs against SENP6 (siSENP6) or a pool of four nontargeting siRNAs (NTP). Cells were fixed and stained with Hoechst to visualize DNA and CENP-T antibody 2 days post transfection. Panels show representative pictures of mitotic (left panel) and interphase cells (right panel). Scatter plots show quantifications of the average CENP-T foci intensities per cell (for mitotic cells) or per picture (for interphase cells). A two-sided $t$-test was performed. ****$p < 0.0001$. $n$ (NTP mitotic cells) = 15; $n$ (siSENP6 mitotic cells) = 16; $n$ (NTP interphase cells) = 16; $n$ (siSENP6 interphase cells) = 16. Dashed lines indicate areas of DNA. **b** U2OS cells stably expressing GFP-CENP-W were treated as in panel **a**. Cells were fixed and stained with Hoechst to visualize DNA and GFP antibody to enhance GFP signal. Left panels show representative pictures of mitotic (left panels) and interphase cells (right panels). Scatter plots show quantifications of the average CENP-W foci intensities per cell (for mitotic cells) or per picture (for interphase cells). A two-sided $t$-test was performed. ****$p < 0.0001$; ***$p < 0.001$. $n$ (NTP mitotic cells) = 15; $n$ (siSENP6 mitotic cells) = 15; $n$ (NTP interphase cells) = 16; $n$ (siSENP6 interphase cells) = 16. **c** U2OS cells were treated as in panel **a**, fixed and stained with Hoechst to visualize DNA and CENP-A antibody 2 days post transfection. Panels show representative pictures of mitotic (left panel) and interphase cells (right panel). Scatter plots show quantifications of the average intensities of CENP-A foci per cell (for mitotic cells) or per picture (for interphase cells). A two-sided $t$-test was performed. ****$p < 0.0001$. $n$ (NTP mitotic cells) = 16, $n$ (siSENP6 mitotic cells) = 16; $n$ (NTP interphase cells) = 16, $n$ (siSENP6 interphase cells) = 16. Dashed lines indicate areas of DNA. Scale bars = 5 μm (mitotic cells), 10 μm (interphase cells). All error bars shown represent standard deviation (SD). Source data are provided as a Source Data file

required for its centromeric localization throughout the cell cycle (Fig. 6a and Supplementary Fig. 5a, b). CENP-W, the direct binding partner of CENP-T, also showed reduced accumulation at centromere foci (Fig. 6b and Supplementary Fig. 5c, d). In conclusion, deSUMOylation of the CCAN subunits CENP-T and CENP-W by SENP6 is important for efficient localization to mitotic and interphase centromeres.

Furthermore, we tested whether poly-SUMOylation of Mis18BP1 and Mis18A upon SENP6 knockdown affects the activity of the MIS18 complex to incorporate CENP-A in centromeres. We found that knockdown of SENP6 reduces the accumulation of CENP-A in centromeres (Fig. 6c and Supplementary Fig. 5e–g). This effect could be rescued by expressing exogenous knockdown-resistant wild-type SENP6, but not by

exogenous knockdown-resistant catalytic dead SENP6 (Fig. 7 and Supplementary Fig. 6a, b). In our rescue experiments of the second shRNA directed against SENP6, we noticed a slight increase and a slight decrease of average CENP-A foci intensities in the two replicates that canceled out each other.

**Subcellular localization of SENP6.** To address whether deSU-MOylation of Mis18BP1 and CCAN subunits occurs at centromeres or in the nucleoplasm prior to accumulation of Mis18BP1 and CCAN subunits at centromeres, we investigated the subcellular localization of SENP6 by fluorescence microscopy. We found that SENP6 does not accumulate at centromeres in mitotic cells or in interphase cells (Fig. 8). Instead SENP6 is located in the nucleoplasm in interphase cells and in a pattern excluded from condensed chromosomes in mitotic cells. It is therefore likely that deSUMOylation of Mis18BP1 and CCAN subunits occurs prior to their accumulation at centromeres.

**Chromatin depletion of CCAN proteins upon SENP6 knockdown.** Since we have shown that SENP6 is responsible for the group deSUMOylation of the CCAN proteins, we wondered if the decreased accumulation of CENP-T and -W at the centromere does also apply to the other CCAN components. Therefore, we isolated chromatin fractions from cells either treated with a nontargeted siRNA pool or a pool of four siRNAs directed against SENP6. Additional to CENP-T, we could identify CENP-C, -K, -Q, -P, -N, -O, and -I to be substantially depleted from the chromatin fraction after SENP6 knockdown. CENP-H, -A, and -U were depleted to a lesser degree (Fig. 9a and Supplementary Data 3). Immunoblot validation of CENP-K, -P/O, -T, and -C confirmed that SENP6 knockdown induced chromatin depletion of these proteins. The depletion of CENP-H and CENP-A from the chromatin was less pronounced, in agreement with the mass spectrometry data (Fig. 9b and Supplementary Data 3).

**A lack of functional SIMs in CCAN subunits.** SUMOylation is involved in the buildup of protein complexes via phase separation[48]. Surprisingly, SUMOylation of CCAN subunits prevents efficient assembly of the complex. Phase separation of PML bodies requires the presence of functional SIMs[48]. We searched for SIMs in CCAN subunits and noticed only a few potential SIMs in CENP-C, -K,-I, and -P (Supplementary Fig. 7a)[49]. We could show that CENP-C, -H, -K, and -T were unable to bind to a recombinant SUMO trimer despite the presence of potential SIMs in CENP-C and -K in contrast to the well-known SUMO polymer binder RNF4 and SENP6 itself (Supplementary Fig. 7b). This is consistent with our finding that SUMOylation does not stimulate CCAN complex formation. Collectively, our data indicate that subunits of all five CCAN subcomplexes are dependent on deSUMOylation by SENP6 to accumulate at the chromatin, which is a prerequisite to form a functional CCAN network and the basis for the KMN network to assemble during mitosis and promote faithful chromosome congression and segregation (Fig. 9c).

## Discussion

In contrast to widespread knowledge on signal transduction by polymeric ubiquitin, signal transduction by polymeric SUMO has remained virtually unexplored. Uncovering around 180 target proteins linked to poly-SUMO2/3 is a major step forward in our understanding of SUMO polymer signaling, since this provides key insight into the cellular processes regulated by SUMO polymers. Here, we focused on the large group of CCAN proteins, which are regulated by SUMO2/3. Immunoblotting experiments

revealed extensive SUMO chains assembled on these targets that appear similar in size to ubiquitin chains. The absence of these chains in the presence of SENP6 indicated their dynamic nature and rapid processing under regular cell culture conditions, leaving only mono- or di-SUMO molecules attached[11,25]. In agreement with the preference of SENP6 for SUMO chains, mono-SUMO2 attached to CENP-T and free di-SUMO2 are processed less efficiently by SENP6 (Fig. 1b and Supplementary Fig. 1c).

We have uncovered an extensive set of target proteins regulated by poly-SUMO2/3, employing knockdown of SENP6, the major protease that removes SUMO chains from target proteins. In the absence of SENP6, SUMO chains accumulated to high levels on at least 180 targets, including multiple highly interconnected protein networks, such as the CCAN, a large group of DNA damage response factors, proteins involved in ribosomal RNA metabolisms or factors which regulate DNA replication. These different functional groups of target proteins are each regulated in a strikingly group-like manner, suggesting localized activity of SENP6 under regular cell culture conditions. The observed co-regulation of the CCAN proteins by SUMO chains is an exciting example of co-regulation of a group of functionally related proteins as initially proposed by Johnson and Blobel for the yeast septins[50] and further developed by Jentsch and Psakhye for yeast proteins involved in DNA repair[10]. A related key question is how these chains are assembled. According to the model proposed by Jentsch and Psakhye, the SIMs in the SUMO E3 ligases play a major role to recruit and stabilize these E3 ligases at the site of an initial SUMOylation event, enabling a wave of SUMOylation in a highly localized manner[10]. Likewise the SIMs which are located in the N terminus of SENP6 could potentially tether the protease to a SUMOylation hub, leading to the deSUMOylation of all proteins in the vicinity (Fig. 9c). The principle of group regulation explains the relatively small number of identified SUMO E3 ligases and SUMO proteases and also provides an explanation for the redundancy of single SUMOylation and deSUMOylation events.

Our data indicate that SENP6 and SENP7 have largely non-overlapping roles, since SENP7 was unable to functionally compensate for the absence of SENP6. This would indicate that the substrates for SENP7 differ from the substrates for SENP6, or that SENP7 is active in different cell types or at different times compared with SENP6. Most interestingly, SENP7 was found to regulate the enrichment of the histone mark H3K9Me3 reader HP1 specifically at the pericentric heterochromatin that is critical for centromere function[51,52]. SENP7 depletion leads to a delocalization of HP1 but can be rescued completely by catalytically dead SENP7, thus the regulatory control of SENP7 of the pericentric heterochromatin is independent of its protease activity[53,54]. However, SENP7 protease activity was shown to counter chromatin condensation upon DNA damage via the deSUMOylation of KAP1/TRIM28 thereby preventing recruitment of chromatin condensation-stimulating remodelers and indirectly leading to chromatin relaxation and proficient DNA damage repair[55].

Recently, SENP6 was also shown to target KAP1/TRIM28 in mouse rib chondrocytes. The failure of deSUMOylation of KAP1/TRIM28 was shown to be responsible for increased p53 activity that led to enhanced senescence and apoptosis of chondrocytes and osteochondroprogenitor cells responsible for the observed premature aging phenotype of mice deficient for SENP6[56]. These findings together with our observations demonstrate that SENP6 and SENP7 are involved in similar cellular pathways, such as centromere integrity and the DNA damage response, but how SENP6 and SENP7 are orchestrated together in these processes needs to be further investigated. Of note, the observed premature aging phenotype in induced SENP6 knockout mice resembles

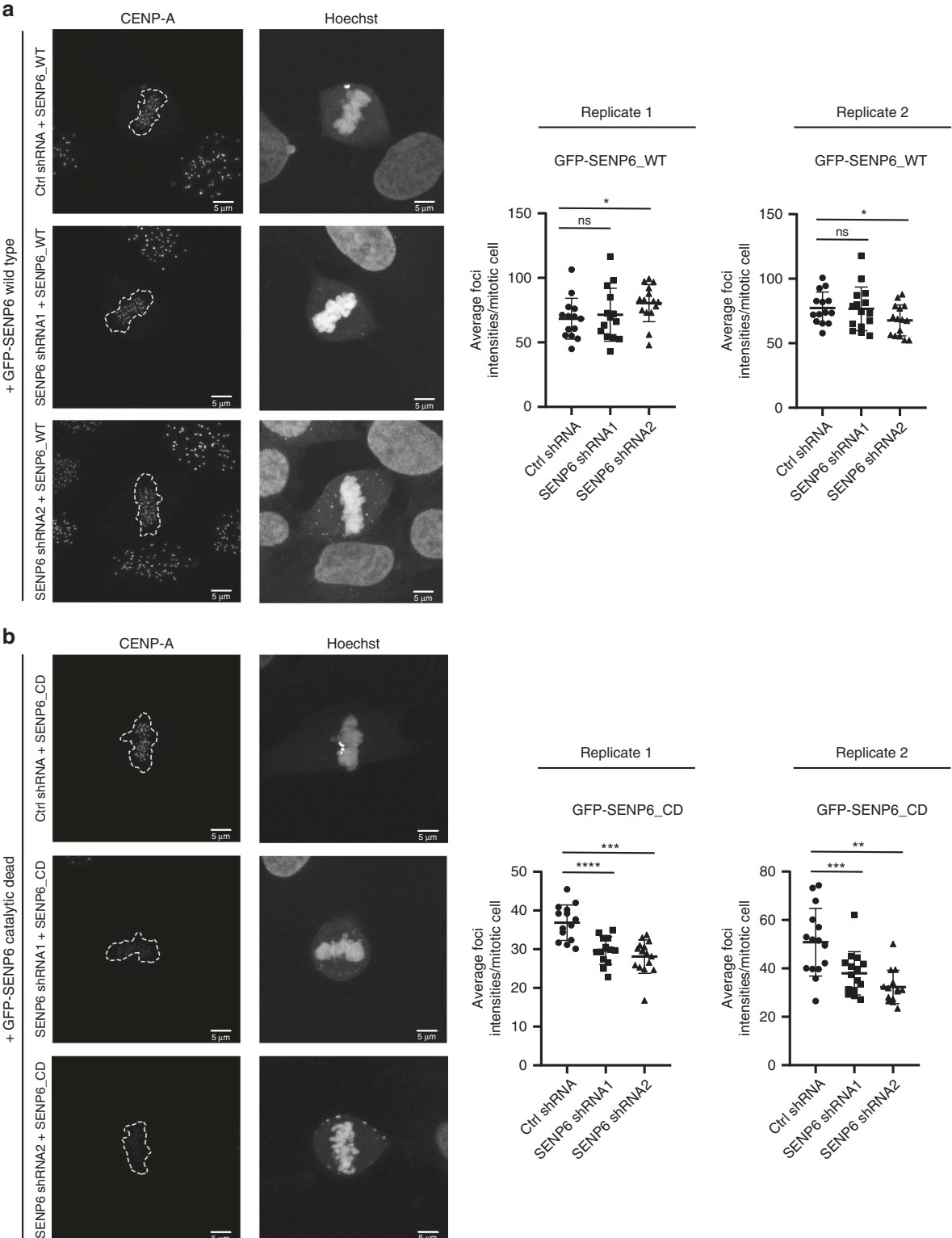

**Fig. 7** CCAN protein localization at the centromere depends on catalytic activity of SENP6. Knockdown of SENP6 can be rescued by reintroduction of wild-type SENP6, but not by reintroduction of catalytic dead SENP6. **a, b** U2OS cells stably expressing inducible shRNA-resistant GFP-tagged wild type (WT) (**a**) or catalytic dead (CD) SENP6 (**b**) were established. Expression of these constructs was induced by doxycycline for 24 h prior to transduction with lentiviruses encoding SENP6 shRNAs. Medium was replaced 1 day post infection. The next day, cells were seeded on coverslips and grown overnight. Subsequently, cells were fixed and stained with Hoechst to visualize DNA and CENP-A antibody. Panels show representative pictures of mitotic cells. Scatter plots show quantifications of the average CENP-A foci intensities per cell for two independent replicates. The data were statically analysed by two-sided *t*-test. *$p < 0.05$; **$p < 0.01$; ***$p < 0.001$, ****$p < 0.0001$. For **a** replicate 1: *n* (ctrl shRNA) = 14 cells, n (SENP6 shRNA1) = 14 cells; n (SENP6 shRNA2) = 15 cells; replicate 2: n (ctrl shRNA) = 14 cells, *n* (SENP6 shRNA1) = 15 cells; *n* (SENP6 shRNA2) = 15 cells. For **b** replicate 1: *n* (ctrl shRNA) = 14 cells, *n* (SENP6 shRNA1) = 13 cells; *n* (SENP6 shRNA2) = 15 cells; replicate 2: *n* (ctrl shRNA) = 15 cells, *n* (SENP6 shRNA1) = 15 cells; *n* (SENP6 shRNA2) = 14 cells. Dashed lines indicate areas of DNA. Scale bars = 5 μm. Error bars represent standard deviation. Source data are provided as a Source Data file

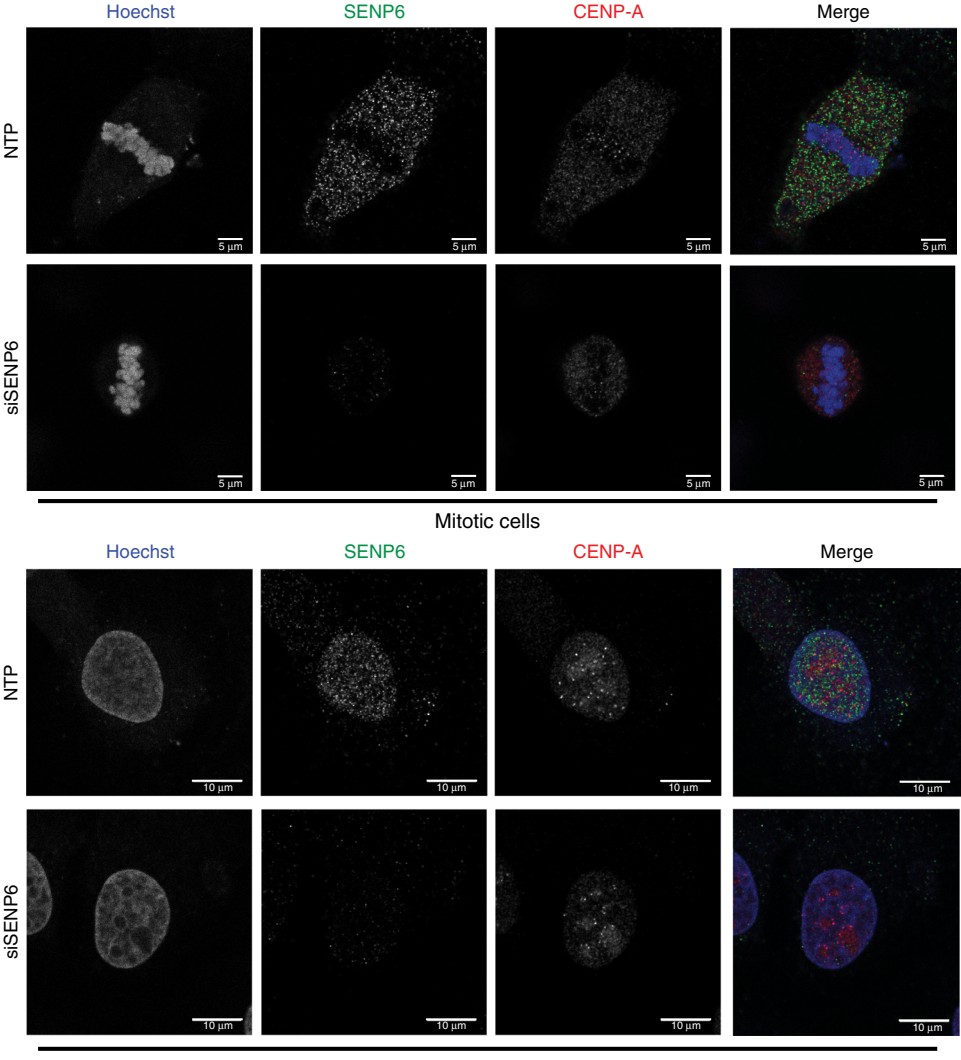

**Fig. 8** SENP6 localizes to the nucleoplasm. U2OS cells were grown on coverslips overnight, transfected either with a pool of four siRNAs against SENP6 (siSENP6) or a pool of four nontargeting siRNAs (NTP). Cells were fixed 2 days post transfection and co-stained with Hoechst to visualize DNA and antibodies directed against SENP6 and CENP-A. Panels show representative pictures of mitotic (upper panel) and interphase cells (lower panel). Scale bar = 5 μm (mitotic cells), 10 μm (interphase cells). Source data are provided as a Source Data file

phenotypes of mice deficient in DNA repair and surveillance proteins[57]. In line with this observation, our project uncovered an extensive set of DNA damage response factors regulated by SUMO chains. This argues for a key regulatory role of SENP6 and poly-SUMO2/3 in the DNA damage response that is much broader than the initially identified RPA70 protein[58].

Poly-SUMO2/3, which accumulated in response to SENP6 knockdown, on CCAN subunits inhibited their accumulation at the centromeres. SENP6 depletion was earlier demonstrated to result in failure of chromosome congression due to RNF4-dependent degradation of CENP-I and subsequently reduced recruitment of the KMN network components Ndc80 and Mis12[30]. Here we show that most CCAN subunits are regulated by SENP6. We hypothesize that the downstream effects on Ndc80 and Mis12 upon SENP6 knockdown are mediated by the global induction of poly-SUMO2/3 on multiple CCAN subunits. We demonstrate a significant SENP6 knockdown-induced reduction of CENP-T and -W foci. These members of the CCAN network make direct contact with Ndc80 and therefore qualify, as much as CENP-I, as initial points of misregulation[46]. The accumulation of CENP-A at centromeres is also reduced, most likely as a result of

poly-SUMOylation of Mis18BP1 and Mis18A in the absence of SENP6. It should be noted that stabilization of CENP-A at centromeres requires the presence of CCAN subunits[59,60].

Recently, it has been shown that SUMOylation can lead to phase separation of proteins, exemplified by the PML-body[48]. In case of the CCAN, we found that extensive SUMOylation does reduce instead of enhance protein complex formation. The main difference between the PML-body and the CCAN appears to be the presence of functional SIMs in PML-body components and the very limited set of potential SIMs in CCAN subunits. Our data thus indicate that SUMO polymers on centromeric proteins have an opposite role to counteract centromere assembly. One possible explanation could be that the bulky accumulated poly-SUMO2/3 chains interfere with the formation of the CCAN at the centromeres by steric hindrance due to interfering with direct binding to other CCAN proteins or to centromeric DNA.

The known function of SUMO chains is their role in protein degradation. SUMO chains provide an efficient binding site for SUMO-targeted ubiquitin ligases including RNF4. Tandem arrays of SIMs in RNF4 enable preferential binding to SUMO chains[35]. Ubiquitination of SUMOylated proteins targets them to the

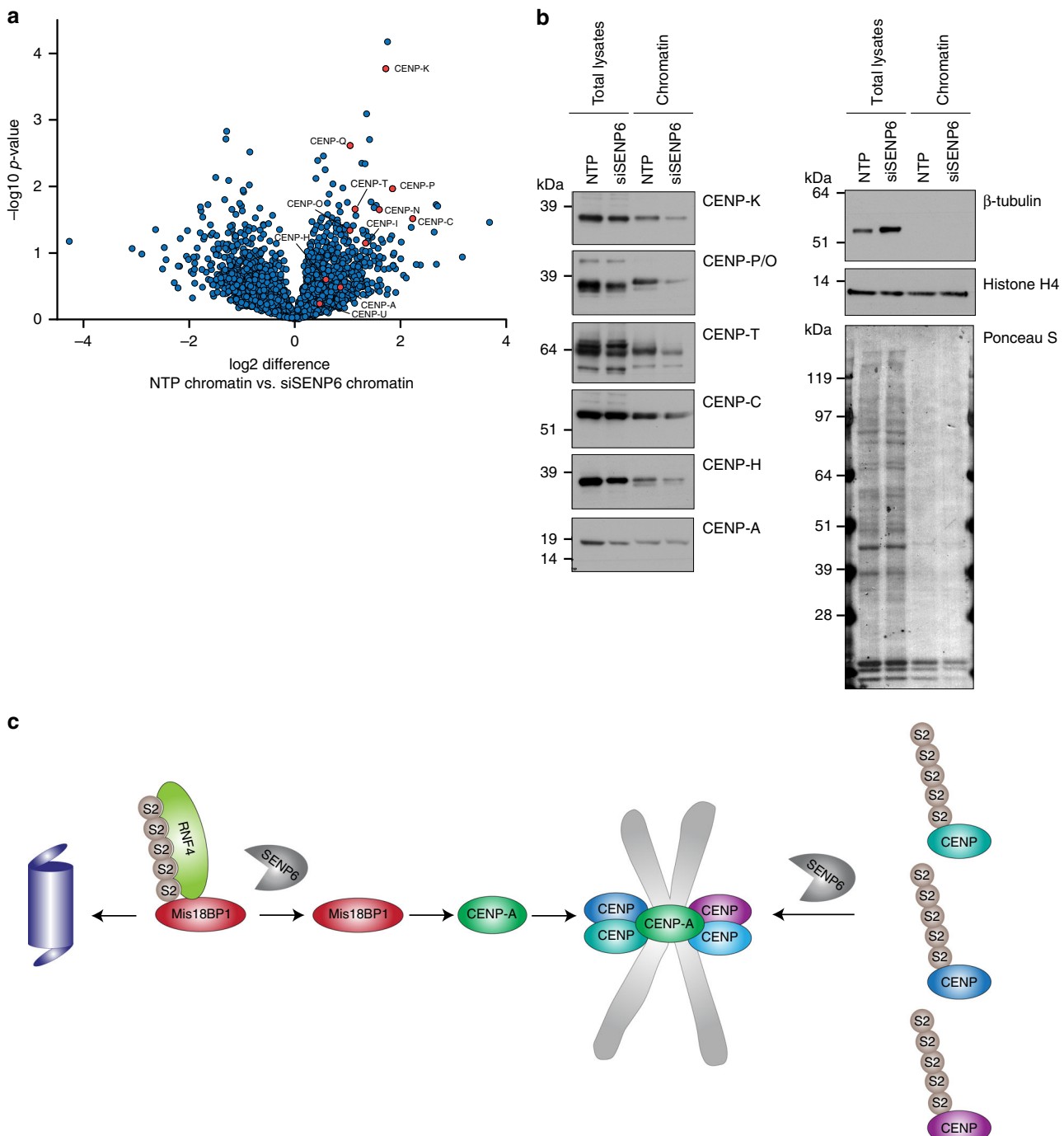

**Fig. 9** SENP6 knockdown leads to the depletion of multiple CCAN proteins from chromatin. **a** U2OS cells were transfected either with a pool of four siRNAs against SENP6 (siSENP6) or a pool of four nontargeting siRNAs (NTP). Chromatin fractions were isolated and proteins were analyzed by mass spectrometry. Volcano plot showing all identified proteins within the NTP treated sample compared with the siSENP6 treated sample. All identified inner kinetochore proteins are represented by red circles. $n = 4$ independent experiments. **b** Samples from panel **a** were analyzed by immunoblotting using antibodies against several inner kinetochore proteins, as well as β-tubulin and histone H4 as control for efficient fractionation. Source data are provided as a Source Data file. **c** Model of SENP6 regulation of CCAN proteins. SENP6 deSUMOylated CCAN proteins as well as Mis18BP1. SENP6 depletion leads to RNF4-mediated degradation of Mis18BP1 and consequently the failure of CENP-A to accumulate at the centromere. Reduced CENP-A levels as well as accumulated SUMO chains on other CCAN family members hinder efficient CCAN protein complex formation at the centromere

proteasome for degradation. The initially identified substrates for this combinatorial PTM pathway are PML and the protein product of the causative fusion gene for acute promyelocytic leukemia, PML-RARα[35,37]. In agreement with this model, we have recently shown that the large majority of proteins that are co-modified by SUMO and ubiquitin accumulated upon proteasome

inhibition[61]. No CCAN subunits were identified in this screen with the only exception of CENP-C.

In addition, we tested this model for centromeric proteins conjugated to poly-SUMO2/3, which accumulated in absence of SENP6, and surprisingly found that proteasome inhibition did not enhance the presence of SUMO chains on these proteins. In

contrast, we mainly noted a reduction in SUMO chain levels attached to centromeric proteins upon proteasome inhibition, most likely due to a reduced pool of free SUMO. We also failed to observe an increase in ubiquitination of the individual CCAN proteins following SENP6 depletion and proteasome inhibition, whereas we noticed a global increase of co-modified proteins as expected. We therefore hypothesize that SUMO chains attached to centromeric proteins are not functioning as a degradation signal. SUMO2/3 chain formation is thought to occur predominantly via lysine 11, which is embedded in a SUMOylation consensus motif, defined by ψKxE motif (where ψ represents hydrophobic amino acids and x represents any amino acid). Site-specific mass spectrometry has however identified different SUMO lysines that mediate polymerization, indicating potential for differential SUMO chain architecture[62]. Whether different SUMO chains mediate different biological fates of target proteins analogous to the ubiquitin system and recruit SENPs, STUbLs or other effector proteins with varying affinities remains to be investigated.

In summary, we provide a resource of target proteins regulated by poly-SUMO2/3 that encompasses multiple protein clusters, including a cluster of centromere-associated proteins and a cluster of DNA damage response factors. Detailed functional analysis of this large set of SUMO polymer targets might widen our understanding on the diverse roles of SUMO polymers in biology. This is analogous to the ever increasing roles uncovered for ubiquitin polymers[63]. We also demonstrate that SENP6 depletion leads to a reduction in cell proliferation and mitotic problems that ultimately lead to cell death, implicating a clinical potential for a SENP6 inhibitor against highly proliferative cancer cells. The recently published SUMO E1 inhibitor also caused mitotic problems, especially the accumulation of anaphase bridges[64,65]. Combined this indicates that precise timing of SUMO conjugation and precise timing of SUMO deconjugation are both essential for proper mitotic progression. Targeting the SUMO conjugation machinery and deconjugation machinery might therefore both be beneficial as anticancer strategies. Unraveling signal transduction by poly-SUMO2/3 might enhance our insight into the regulatory consequences of SUMO polymers and could potentially be employed to reduce cancer cell proliferation.

## Methods

**Cell culture and generation of cell lines**. U2OS cells (ATCC® HTB-96™) (gender: female), HeLa cells (EMBL) (gender: female), HEK 293T[66], and HEK 293GP[67] cells were cultured in Dulbecco's modified Eagle's medium (DMEM) (high glucose, pyruvate, Gibco™) supplemented with 10 % FCS and 100 U/ml penicillin and 100 μg/ml streptomycin (Gibco™). For induction of PML ubiquitination by RNF4, cells were treated with As$_2$O$_3$ (Sigma-Aldrich, 17971) at 1 μM for 4 h.

**In vitro SUMOylation and deSUMOylation assay**. Recombinant human CENP-T-FLAG or human CENP-T-HA were expressed in U2OS cells and purified using Anti-FLAG M2 agarose affinity gel (Sigma-Aldrich, A2220) or EZview Red Anti-HA Affinity Gel (Sigma-Aldrich, E6779), respectively. The purified recombinant proteins were eluted with 150 ng/μl FLAG (custom made) or HA (Sigma-Aldrich, I2149) peptide in 50 mM HEPES pH 7.4, 150 mM KCl, 10 % glycerol. In vitro SUMOylation was carried out by incubating 150 ng recombinant CENPT-FLAG or CENP-T-HA with 500 ng SAE1/2, 2.0 μg UBC9, 2.0 μg SUMO-2, 50 mM Tris pH 7.5, 5 mM MgCl$_2$, 2 mM ATP, 3.5 U/μl creatine kinase, 10 mM creatine phosphate, 0.6 U/μl inorganic pyrophosphate in 20 μl reaction at 37 °C for 3 h. In addition CENP-T-HA was also SUMOylated with SUMO-2 lysine-deficient mutant (K0). As a negative control, recombinant CENP-T-FLAG and CENP-T-HA were treated similarly but leaving out SAE1/2 and SUMO-2 respectively from the reaction. For SENP6 deconjugation assay 10 μl of SUMOylated CENP-T-FLAG was incubated with 150 nM recombinant human SENP6 catalytic domain in buffer containing 25 mM Tris pH 8.0, 150 mM NaCl. 0.1 % Tween-20, 2 mM dithiothreitol (DTT) at 37 °C for 2 h and the reaction was stopped by adding NuPage LDS (4X) sample buffer. Plasmids used for the production of recombinant CENPT-FLAG, CENPT-HA, SAE1/2, UBC9, SUMO2 WT, SUMO-2 K0, and SENP6 catalytic domain are listed in Supplementary Table 2.

**Lentivirus production and transduction**. For lentiviral production of shRNAs, HEK 293T cells were transfected with lentiviral packaging plasmids and plasmids containing SENP6 or RNF4 targeting shRNA or nontargeting control shRNA. Lentivirus was harvested in DMEM medium. For shRNA mediated knockdown experiments, cells were infected with a multiplicity of infection (MOI) of three with third generation lentiviruses encoding SENP6 or RNF4. Transductions were performed in DMEM containing 8 μg/ml polybrene. Medium was replaced after 24 h of infection. Cells were lysed three days post infection. Plasmids for lentivirus production and targeted shRNAs as well as the nontargeting control shRNA SHC002 are derived from Mission shRNA library (Sigma) and are listed in Supplementary Table 2.

**Retrovirus production and transduction**. For stable expression of human GFP-CENP-W, HEK 293GP cells were transfected with pBabe-puro-GFP-CENP-W together with a plasmid encoding the viral envelop VSV-G protein. Retrovirus was harvested and transduction of U2OS and HeLa cells was performed in DMEM containing 8 μg/ml polybrene. Cells expressing GFP-CENP-W were selected with 1 μg/ml puromycin supplemented DMEM medium. Plasmids for retroviral production are listed in Supplementary Table 2.

**SiRNA-mediated knockdown**. For siRNA mediated knockdown of SENP6, 10 nM SMARTpool ON-TARGETplus SENP6 siRNA, or SMARTpool ON-TARGETplus nontargeting siRNA (Dharmacon) was mixed with Opti-MEM™ (Gibco) and Lipofectamine RNAiMAX (Thermo Fischer Scientific, 13778075) before transfection of U2OS cells. Cells were processed 48 h after transfection. siRNA mediated knockdown of CENP-A was performed using the same transfection protocol with 10 nM and 50 nM of SMARTpool ON-TARGETplus CENP-A siRNA (Dharmacon). Cells were processed 48 h after the final transfection. siRNAs used for SENP6 and CENP-A knockdown are listed in Supplementary Table 2.

**Colony formation assay**. For colony formation assays, batches of 1 million U2OS cells were each infected with an MOI of three with lentiviruses encoding shRNA against SENP6. After 1 day of infection 2500 cells were seeded for each condition onto 10 cm diameter dishes in *triplo*. Cell were fixed on day 14 post infection with 100 % methanol for 30 min at −20 °C. Colonies were stained with 0.05 % crystal violet solution for 20 min. before rinsing of the plates with water. Subsequently, crystal violet was solubilized with 1.5 ml MetOH for 10 min and absorbance was measured at 595 nM.

**Rescue experiments**. U2OS cells stably expressing inducible wild type or catalytically dead GFP-SENP6 fusion constructs were established. These constructs carried silent mutations to make them resistant against shRNA-mediated knockdown. Oligonucleotides used to introduce the mutations in SENP6 are listed in Supplementary Table 2. Cells were seeded in 10 cm plates at a density of $0.8 \times 10^6$ cells. The next day, expression of the GFP-SENP6 fusion constructs was induced with doxycycline at 1 μg/ml. The following day, cells were infected with third generation lentiviruses encoding shRNAs targeting SENP6 or control at MOI of three. Medium was refreshed the following day. The next day, cells were seeded on coverslips for microscopy and in 12-well plates for lysis. After overnight incubation, cells were fixed and processed for microscopy or lysed and processed for immunoblotting.

**Fluorescent-activated cell sorting**. Assessment of cell cycle progression was essentially performed as previously described[68] with minor modifications. In brief, HeLa cells were harvested by trypsinization washed once in PBS and resuspended in 1 ml of PBS. Four ml of 100% ethanol were added and the cells were fixed at 4 °C overnight. On the day of flow cytometry analysis, the cells were first centrifuged at $500 \times g$ for 2 min, the supernatant was removed and the cells were washed with PBS and 2% calf serum. Then, the cells were pelleted again and resuspended in 500 μl of PBS complemented with 2% calf serum, 25 μg/ml propidium iodide (Sigma-Aldrich, P4170) and 100 μg/ml RNAse A (Sigma-Aldrich, R6513). Cellular DNA content was determined by flow cytometry with the BD LSRII system and BD FACS DIVA Software (BD Biosciences Clontech). Cell cycle analysis was performed with FlowJo version 10 software using the Watson univariate model. Samples were included in analysis when the values of coefficient of variance (CV) of G1 and G2/M peaks were below 5.

**His10-SUMO2 and His10-ubiquitin purification**. His10-SUMO2 conjugates were purified as previously described[38]. In brief, U2OS cells stably expressing His10-SUMO2 were lysed in 25 pellet volumes of 6 M Guanidine-HCL, 100 mM sodium phosphate, 10 mM Tris, buffered at a pH of 8.0. Lysates were subsequently snap-frozen and stored at −80 °C until further processing. Lysates were thawn at room temperature (RT), sonicated 2× for 5 s at 30 W and subsequently supplemented with 5 mM β-mercaptoethanol and 50 mM imidazole pH 8.0. Prewashed Ni-NTA beads (Qiagen, 30210) were added to the lysates and incubated for 3–5 h at RT or overnight at 4 °C. Ni-NTA beads were washed with wash buffer 1–4, respectively; Wash buffer 1: 6 M Guanidine-HCL, 100 mM sodium phosphate, 10 mM Tris, 10 mM imidazole, 5 mM β-mercaptoethanol, 0.2% Triton X-100. Wash buffer

2: 8 M urea, 100 mM sodium phosphate, 10 mM Tris, 10 mM imidazole, 5 mM β-mercaptoethanol, 0.2% Triton X-100. Wash buffer 3: 8 M sodium phosphate, 10 mM Tris, 10 mM imidazole, 5 mM β-mercaptoethanol, 0.2% Triton X-100. Wash buffer 4: 8 M urea, 100 mM sodium phosphate, 10 mM Tris, 5 mM β-mercaptoethanol, 0.1% Triton X-100. For samples used for subsequent mass spectrometry analysis, 0.2% Triton X-100 was included in Wash 1 and 0.1% Triton X-100 was included in Wash 2, Wash 3, and Wash 4 did not contain Triton X-100. Purified proteins were twice eluted in one bead volume of 7 M urea, 100 mM sodium phosphate, 10 mM Tris, and 500 mM imidazole pH 7.0.

**Chromatin fractionation.** Cells were harvested washed twice in ice-cold PBS. A small fraction of cell suspension was lysed in SNTBS (2% SDS, 1% NP-40, 50 mM Tris pH 7.5, 150 mM NaCl) buffer as input control. The residual cells were lysed in buffer A (10 mM HEPES, 10 mM KCL, 1.5 mM $MgCl_2$, 10% glycerol, 340 mM sacharose, 1 mM DTT, 1 protease inhibitor tablet without EDTA (Complete Mini protease inhibitor cocktail, Sigma-Aldrich, 11836170001)/10 ml. The lysate was subsequently supplemented with Triton X-100 to a final concentration of 0.1 %. The lysate was incubated on ice for 8 min and subsequently centrifuged at $1300 \times g$ for 5 min at 4 °C. Supernatant was collected as cytoplasmic fraction. Pellet was washed twice with buffer A and subsequently lysed in buffer B (3 mM EDTA, 0.2 mM EGTA, 1 mM DTT, 1 protease inhibitor tablet without EDTA/10 ml) for 30 min at 4 °C. After centrifugation at $1700 \times g$ for 5 min at 4 °C supernatant (nucleoplasmic fraction) was separated from pellets (chromatin fraction). The chromatin fraction was further diluted in 100 μl SNTBS buffer and heated to 99 °C for 10 min.

**SUMO Q87R in vitro SUMOylation of CENPT.** Recombinant CENP-T-HA was SUMOylated in vitro by adding 50 mM Tris pH 7.5, 5 mM $MgCl_2$, 2 mM ATP, 3.5 U/ml Creatine kinase, 10 mM creatine phosphate, 0.6 U/ml inorganic pyrophosphate, 5 μg SAE1/2, 20 μg UBC9, and 20 μg FLAG-SUMO-2-Q87R to 2 μg samples of CENPT-HA in a volume of 200 μl and incubating for 3 h at 37 °C. As a negative control, recombinant CENPT-HA was treated similarly but leaving out FLAG-SUMO-2-Q87R from the reaction. For mass spectrometry analysis, SUMOylated CENP-T-HA was incubated with 50 μl prewashed EZview Red Anti-HA Affinity Gel (Sigma-Aldrich, E6779) for 2 h at 4 °C in 50 mM Tris pH 7.5 and 150 mM NaCl. HA beads were washed with 50 mM Tris pH 7.5, 150 mM NaCl, and 20 mM NEM to eliminate the SUMO machinery. Subsequently, beads were washed three times with 50 mM ammonium bicarbonate (ABC) and subsequently incubated with 2 μg of trypsin (Promega, V5111) overnight at 37 °C. The samples were passed through a prewashed 0.45 μm filter (Millipore) to remove the HA beads and acidified with 2% trifluoroacetic acid (TFA) (Sigma). Peptides were desalted and concentrated on triple-disc C18 reverse phase Stage Tips. Peptides were eluted with acetonitrile (ACN), vacuum dried and dissolved in 0.1% folic acid. Plasmids used for the production of recombinant CENPT-HA, FLAG-SUMO-2-Q87R, SAE1/2, and UBC9 are listed in Supplementary Table 2.

**In solution digestion and stage tipping.** His10-SUMO2 purified elutions were concentrated using a 100 kD cutoff filter and diluted with ABC to an end concentration of 50 mM. Samples were reduced with DTT in two steps, first to 1 mM DTT and subsequently to 6 mM DTT. In between the reduction steps, sample was alkylated using 5 mM chloroacetamide. Proteins were first digested with Lys-C (Promega, VA1170) in a 1:100 enzyme-to-protein ratio for 5 h. Peptides were diluted with 50 mM ABC before trypsin (Promega, V5111) digestion. Trypsin digestion was carried out in a 1:100 enzyme-to-protein ratio, overnight and in the dark at RT. After digestion peptides were acidified with 2% TFA and then desalted and concentrated on triple-disc C18 reverse phase StageTips[69]. Peptides were eluted with ACN, vacuum dried and dissolved in 0.1% folic acid.

**In gel digestion.** Chromatin fractions in SNTBS buffer were loaded onto a precast 4–12% Bis-Tris gel (Bold, Thermo Fischer Scientific). Proteins were excised from the gel, divided into two fractions (high and low molecular weight bands) and cut into small $1 \times 1$ mm cubes. Gel pieces were destained with 25 mM ABC in 50% ACN twice for 20 min at 15 °C. Gel pieces were dehydrated with 100% ACN for 10 min at 25 °C and subsequently vacuum dried. Gel pieces were rehydrated and proteins were reduced with 10 mM DTT in 50 mM ABC and incubated for 60 min at 56 °C. Gel pieces were subsequently alkylated with 55 mM iodoacetamide in 50 mM ABC for 45 min at 25 °C. Gel pieces were subsequently washed twice with 50 mM ABC followed by dehydration with 100% ACN. Gel pieces were vacuum dried and rehydrated with 12.5 ng/μl trypsin (Promega, V5111) in ABC overnight. After acidifying the gel pieces with 100% TFA, peptides were extracted twice from gel pieces with 3 % TFA in 30% ACN followed by dehydration with 100% ACN. Peptides were vacuum dried to remove ACN, acidified with 2% TFA and subsequently desalted and concentrated on triple-disc C18 reverse phase Stage Tips. Peptides were eluted with ACN, vacuum dried and dissolved in 0.1% folic acid.

**Electrophoresis and immunoblotting.** Proteins were separated on either precast 4–12 % Bis-Tris gradient gels (Bold, Thermo Fischer Scientific) or precast 3–8% tris-acetate gels (NuPage, Thermo Fischer Scientifc). Separated proteins were subsequently transferred to Amersham Protran Premium 0.45 μm nitrocellulose

membranes (Sigma-Aldrich) using a submarine system. Membranes were stained with Ponceau S solution for visualization of total protein content and blocked with PBS containing 8% milk powder and 0.05% Tween-20 before incubating with the primary antibodies. Primary antibodies were diluted in 8% milk, 0.05% Tween-20, 1x PBS and incubated with membranes overnight at 4 °C. Primary antibodies and dilutions used are listed in Supplementary Table 1. Donkey anti-rabbit IgG-HRP and goat anti-mouse IgG-HRP were used as secondary antibodies 1:5000 dilution in 8% milk. Signal was detected and captured by using Pierce ECL2 (Life technologies) and RX medical film (Fuji).

**LC–MS/MS analysis.** Vacuum-dried peptides were resuspended in 0.1% formic acid (FA) prior to liquid chromatography-tandem mass spectrometry. All analyses were performed on an EASY-nLC 1000 system (Proxeon, Odense, Denmark) connected to a Q-Exactive Orbitrap (Thermo Fisher Scientific, Germany) through a nano-electrospray ion source. Separation of peptides was achieved using a 15 cm analytic column with an inner diameter of 75 μm, packed in-house with 1.9 C18-AQ beads. For the identification of SENP6-regulated proteins, peptides were analyzed over a 120 min gradient from 2 to 95% ACN in 0.1% FA. The mass spectrometer was operated in data-dependent acquisition mode using a top 10 method. Full-scan MS spectra acquired at a target value of 3E6 and a resolution of 70,000. The higher-collisional dissociation (HCD) tandem MS/MS were acquired using a target value of 1E5, a resolution of 17,500 and a normalized collision energy (NCE) of 25%. The maximum injection times for MS1 and MS2 were 20 and 60 ms, respectively. For the identification of chromatin-associated proteins upon SENP6 depletion, peptides were analyzed over a 4-hour gradient from 2 to 95% CAN in 0.1% FA. The mass spectrometer was operated in data-dependent acquisition mode using a top 7 method. Full-scan MS spectra acquired at a target value of 3E6 and a resolution of 70,000. The HCD tandem MS/MS were acquired using a target value of 1E5, a resolution of 35,000 and an NCE of 25%. The maximum injection times for MS1 and MS2 were 50 and 120 ms, respectively.

**MaxQuant data analysis.** For the analysis of SENP6-regulated SUMO proteins, four experimental conditions were performed in biological triplicate and each sample was measured with two technical repeats, which resulted in a total of 24 MS runs. All RAW data were analyzed using MaxQuant software version 1.5.3.30[41] and its integrated search engine Andromeda. The search was performed against an in silico digested reference proteome for Homo Sapiens obtained from Uniprot.org (June 24th 2016). Database searches were performed with trypsin and Lys-C allowing two missed cleavages. Carbamidomethyl was set as fixed modification and the variable modifications of oxidation (M) and acetyl (protein N-term) were allowed with a max number of 5 modifications per peptide. Fast label-free quantification (LFQ) was enabled with an LFQ minimal ratio count of two, an LFQ minimal number of neighbors of three and an LFQ average number of neighbors of six. Match-between runs was enabled with a match time window of 0.7 min and an alignment time window of 20 min. A maximum peptide mass of 4600 Da was permitted. A first search with a peptide tolerance of 20 ppm was performed to determine a mass and time recalibration. A second search with a peptide tolerance of 4.5 ppm was performed as main search and the results were used for further analysis. Desired false discovery rates (FDRs) for peptide spectrum match (PSM) and protein level were set to 1%. A minimal score for modified peptides of 40 was applied, together with a minimal delta score for modified peptides of six. For protein identification the minimum number of razor + unique peptide was set to one, as was the minimum number of peptides.

For the identification of chromatin-associated proteins upon SENP6 depletions, four experimental conditions (1: nontargeting siRNA low molecular weight bands, 2: nontargeting siRNA high-molecular weight bands, 3: SENP6-targeting siRNA low molecular weight bands, 4: SENP6-targeting siRNA high molecular weight bands), were collected in four biological replicates and measured with one technical repeat, resulting in a total of 16 MS runs. All RAW data were analyzed using MaxQuant software version 1.5.3.30[41] and its integrated search engine Andromeda. The search was performed against an in silico digested reference proteome for Homo Sapiens obtained from Uniprot.org (June 24th 2016). Database searches were performed with trypsin/P allowing two missed cleavages. Carbamidomethyl was set as fixed modification and the variable modifications of oxidation (M) and acetyl (protein N-term) were allowed with a max number of five modifications per peptide. LFQ was enabled with an LFQ minimal ratio count of two. Match-between runs was enabled with a match time window of 0.7 min and an alignment time window of 20 min. A maximum peptide mass of 4600 Da was permitted. A first search with a peptide tolerance of 20 ppm was performed to determine a mass and time recalibration. A second search with a peptide tolerance of 4.5 ppm was performed as main search and the results were used for further analysis. Desired FDRs for PSM and protein level were set to 1%. A minimal score for modified peptides of 40 was applied, together with a minimal delta score for modified peptides of six. For protein identification the minimum number of razor + unique peptide was set to one, as was the minimum number of peptides.

For the SUMO chain detection on in vitro SUMOylated CENP-T, the search was performed against an in silico digested Uniprot reference proteome for Homo Sapiens obtained from Uniprot.org (March 24th 2016). Database searches were performed with trypsin/P allowing three missed cleavages. Maximum number of modifications per peptide was set to three, with the following variable

modifications: Carbamidomethyl (default), protein N-terminal acetylation (default) methionine oxidation (default), QQTGG modification on lysine and to facilitate the detection of pyroQQTGG (PyroQ) remnants on lysines, pyroQQTGG settings were imported into the Andromeda search engine[70]. Fast LFQ was enabled with an LFQ minimal ratio count of two, an LFQ minimal number of neighbors of three and an LFQ average number of neighbors of six. Match-between runs was enabled with a match time window of 0.7 min and an alignment time window of 20 min. A maximum peptide mass of 6000 Da was permitted. A first search with a peptide tolerance of 20 ppm was performed to determine a mass and time recalibration. A second search with a peptide tolerance of 4.5 ppm was performed as main search and the results were used for further analysis. Desired FDRs for PSM and protein level were set to 1%. A minimal score for modified peptides of 40 was applied, together with a minimal delta score for modified peptides of six. For protein identification the minimum number of razor + unique peptide was set to one, as was the minimum number of peptides.

**Perseus data analysis**. For identification of SENP6-regulated SUMOylated proteins, MaxQuant 'protein groups' output tables were subsequently filtered and statistically analyzed using the software package Perseus, version 1.5.0.31[42]. Proteins with potential incorrect identifications ('only identified by site' or 'reverse') were removed before the LFQ intensities were log2 transformed. Replicates of the same condition were grouped together. The data were filtered for protein groups, which had at least three valid values in at least one group. Missing values were replaced by imputation using normally distributed values based on the total data matrix with a randomized 0.3 (log2) width and a 1.8 (log2) down shift. To obtain p values and log2 differences of the protein LFQ intensities in different conditions, a series of two-sided two samples t-tests were performed. Within the Supplementary Data 1 and 2, the header of each column for p values and log2 differences of LFQ intensities indicate the compared conditions. Putative SENP6 regulated SUMO targets were selected based on following criteria: (1) The protein LFQ intensity has a log2 difference of at least one (twofold change) with a −log10 p value of at least 1.3 ($p < 0.05$) in any of the His10-SUMO2 expressing conditions tested (two-sided t-test) against the parental condition (no His10-SUMO2 expression). (2) The protein LFQ intensity has a log2 difference of at least one with a −log10 p-value of at least 1.3 in both SENP6 knockdown conditions when tested (two-sided t-test) against the His10-SUMO expressing nontargeting control condition.

For the analysis of chromatin-associated proteins upon SENP6 knockdown, MaxQuant 'protein groups' output tables were subsequently filtered and statistically analyzed using the software package Perseus, version 1.5.2.4[42]. Proteins with potential incorrect identifications ('only identified by site' or 'reverse') were removed before the LFQ intensities were log2 transformed. Replicates of the same condition were grouped together. The data were filtered for protein groups, which had at least three valid values in at least one group. Missing values were replaced by imputation using normally distributed values based on the total data matrix with a randomized 0.3 (log2) width and a 1.8 (log2) down shift. To obtain p values and log2 differences of the protein LFQ intensities in different conditions, a series of two-sided two samples t-tests were performed. In Supplementary Data 3, the header of each column for p values and log2 differences of LFQ intensities indicate the compared conditions. Putative SENP6-regulated chromatin components were selected based on the following criterium: the protein LFQ intensity has a log2 difference of at least one (twofold change) with a −log10 p value of at least 1.3 ($p < 0.05$) in the SENP6 knockdown conditions tested (two-sided t-test) against the nontargeting control knockdown condition.

**Gene ontology and STRING network analysis**. GO analysis was performed using the GO consortium web tool (www.geneontology.org). For the evaluation of enriched GO terms of the identified SENP6-regulated proteins the PATHER overrepresentation test (released 20171205) was used. The proteins were analysed for overrepresentation of PANTHER GO-Slim biological process, PANTHER GO-Slim cellular component, and PANTHER GO-Slim molecular function terms using the Fischer exact test.

Network analysis of SENP6 regulated SUMO targets (Supplementary Data 2) was performed using the online STRING database v10.5 (www.String-db.org)[71]. The following setting were applied: Output settings: high confidence interaction score (0.7), edges show protein connections based on textmining, experiments, databases, co-expression, neighborhood, co-occurrence, and gene fusion. The network was subsequently exported as TVS (tab separated values) file and imported into Cytoscape version 3.6.1 for further visualization and network analysis. The cytoscape plug-in MCODE (molecular complex detection) version 1.5.1 was used to identify highly connected subclusters of proteins using a degree cutoff of two, cluster finding: haircut, a node score cutoff of 0.2, a K-core of two and a max depth of 100.

**Immunostaining for microscopy**. For immunostainings with CENP-T, GFP, CENP-A, and SENP6 antibody, cells were grown on coverslips and fixed for 15 min with 4% paraformaldehyde in PBS, briefly rinsed with PBS and permeabilized for 15 min at RT with 1x PBS containing 0.05% Triton X-100. Cells were blocked for 15 min at RT in 0.1 M Tris-HCL pH 7.5, 0.15 M NaCl, 5 mg/ml Boehringer Blocking Reagent (TNB) and incubated with primary antibodies diluted in TNB for

1 h. Cells were washed four times with PBS containing 0.05% Tween-20. Secondary antibodies were diluted 1:500 in TNB and incubated for 30 min. Coverslips were washed 4× with 0.05% Tween, 1x PBS and DNA was visualized using 10 mg/ml Hoechst diluted in 0.05% Tween, 1x PBS. Coverslips were subsequently mounted onto microscopy slides using CitiFluor mounting medium (Science Services, E17970-25). Antibodies and dilutions are listed in Supplementary Table 1.

**Imaging and Image analysis**. Images were acquired using a Leica SP8 confocal microscope. For foci quantification of mitotic cells, ~15-20 z-stacks were acquired at 0.2 μm steps and a final pixel size of 45 nm using a ×63 objective, 1.4 NA. For foci quantification of interphase cells ~6-12 z-stacks were acquired at 0.2 μm steps and a final pixel size of 90 nm using a ×63, 1.4 NA objective. Data were analyzed using Image J (Fiji) software. For the analysis of centromere foci, the area of the DNA was selected based on the Hoechst staining. Subsequently, centromere foci were identified by localization of local maxima using the find maxima function within the preselected area. The intensity of each foci was measured for 15 pictures per condition and biological replicate. Statistical significance was determined by two-sided t-tests. $^*p < 0.05$; $^{**}p < 0.01$; $^{***}p < 0.001$, $^{****}p < 0.0001$.

**Purification of His10-SUMO-trimer binding proteins**. Purification of His10-SUMO-trimer binding proteins was essentially performed as described before[64]. In brief, BL21 competent *Escherichia coli* cells (New England Biolabs, Cat#C2527I) were transformed with pHIS-TEV30a:His10-ΔN11-SUMO2-trimer. When the bacterial culture reached an OD600 of 0.6, recombinant protein expression was induced overnight at 25 °C with 0.5 mM IPTG. Subsequently, cells were lysed in 50 mM HEPES pH7.6, 25 mM MgCl$_2$, 0.5 M NaCl, 20% glycerol, 0.1% N-P40, 50 mM imidazole, 1 mM phenylmethanesulfonylfluoride (PMSF), and protease inhibitor cocktail without EDTA (Complete Mini protease inhibitor cocktail, Sigma-Aldrich, 11836170001). The His10-SUMO2-trimer was purified from lysate by incubation with Ni-NTA beads for 3 h at 4 °C. Beads were washed twice in lysis buffer including PMSF and protease inhibitor cocktail and twice in lysis buffer without PMSF and protease inhibitor cocktail. The His10-SUMO2-trimer was eluted with lysis buffer plus 500 mM imidazole for 10 min at 4 °C. The elution step was repeated three times. For the binding assay, His10-SUMO2-trimer was rebound to Ni-NTA beads. Five 15-cm dishes of U2OS cells per sample were lysed in 1 ml of 50 mM Tris pH 7.5, 150 mM NaCl, 0.5 % NP-40, 50 mM imidazole, sonicated and centrifuged at $20,000 \times g$ for 1 h at 4 °C. The supernatant was incubated with recombinant His10-SUMO2-trimer bound to Ni-NTA beads for 2 h at 4 °C. As a control, U2OS lysates were incubated with Ni-NTA beads without His10-SUMO2-trimer. Samples were washed three times with 50 mM Tris pH 7.5, 150 mM NaCl, 0.5 % NP-40, 50 mM imidazole and three times with 50 mM Tris pH 7.5, 150 mM NaCl including tube changes. Binding partners of His10-SUMO2-trimer were eluted with 8 M urea in 50 mM Tris pH 7.5 for 30 min at RT and analysed by immunoblotting. Plasmids used are listed in Supplementary Table 2.

**Statistics**. Unless otherwise indicated, experiments were performed in biological triplicate. Results are presented as mean ± standard deviation. For FACS cell cycle progression analysis, p values were determined by multiple t-tests and corrected for multiple testing by applying the FDR by Benjamini, Krieger, and Yekutieni. For mass spectrometry data, p values were determined using two-sided t-tests. For microscopy analysis of foci intensities p values were determined using two-sided t-test. ($^*p < 0.05$, $^{**}p < 0.01$, $^{***}p < 0.001$, $^{****}p < 0.0001$).

**Reporting summary**. Further information on research design is available in the Nature Research Reporting Summary linked to this article.

## Data availability
The mass spectrometry proteomics data have been deposited to the ProteomeXchange Consortium via the PRIDE partner repository with the dataset identifier PXD011963. The source data underlying Figs. 1b–f, 2b, c, 3a–f, 4a–c, 5a, b, 6a–c, 7a, b, and 9a, b and Supplementary Figs. 1a–d, 3a, b, 4a, b, 5a–g, 6a, b, and 7b are provided as a Source Data file. All other data are available from the corresponding author on reasonable request.

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

## Acknowledgements

We thank Dr R. González-Prieto for assistance with mass spectrometry and Dr L. Voortman for advice on microscopy data analysis. We thank Drs W.C. Earnshaw, P.T. Stukenberg and R.T. Hay for providing antibodies and we acknowledge Dr I.M. Cheeseman for insightful discussion and for providing antibodies. The laboratory of A.C.O.V. is supported by the European Research Council (ERC) and the Netherlands Organisation for Scientific Research (NWO).

## Author contributions

A.C.O.V. conceived and supervised the project. F.L., S.K., E.G. and A.C.O.V. designed experiments. F.L., N.S.J., S.K., E.G., L.A.C., M.V. and E.W. conducted experiments. F.L., N.S.J., S.K. and E.G. analyzed data. F.L. and A.C.O.V. wrote the manuscript. All authors edited the manuscript.

## Additional information

**Competing interests:** The authors declare no competing interests.

**Peer Review Information:** *Nature Communications* thanks Francis Impens and the anonymous reviewer(s) for their contribution to the peer review of this work. Peer reviewer reports are available

