## [Peer Review File · Nature Communications]

Reviewers' comments:

Reviewer #1 (Remarks to the Author):

In this manuscript the authors report the results of a proteomics study designed to identify substrates of the SUMO-specific isopeptidase SENP6. Among their top hits they find subunits of the centromeric CCAN complex, in addition to factors involved in genome maintenance. Their data demonstrate that loss of SENP6 causes a delocalization of CCAN subunits to various different degrees, but that they are apparently not subject to RNF4-mediated ubiquitylation and targeting to the proteasome. From their results the authors conclude that the CCAN complex undergoes a process of group sumoylation/desumoylation and that group desumoylation of the complex by SENP6 is required for correct centromere localization. They thus postulate an alternative, non-degradative function of the poly-SUMO chains in preventing CCAN assembly.

The study is interesting in the sense that most polysumoylation events have been linked to RNF4-initiated degradation, whereas here the authors show that the CCAN subunits are apparently not targeted via this route. Otherwise, the study confirms previous data on the role of SENP6 in centromere assembly and mitosis, but it falls short of proving the authors' main claims: that it is the sumoylation of the CCAN subunits themselves that prevents them from engaging at the centromere. My specific points are as follows:

1. The Dasso lab has already shown that CCAN subunits don't correctly localize to the centromeres in SENP6-depleted conditions (Mukophadhyay, 2006).
2. The conclusion that the CCAN subunits are the relevant SUMO targets responsible for this phenotype is plausible, but the authors' experiments don't provide evidence for this. They exclusively work with SENP6 knockdowns and therefore cannot exclude the alternative model that sumoylation of a different SENP6 substrate is responsible for preventing centromere assembly of the CCAN complex. In fact, a publication that came out very recently in Cell Research (Fu et al. 2019: SENP6-mediated M18BP1 deSUMOylation regulates CENP-A centromeric localization) appears to argue along those lines. If that were the case, the hypersumoylation of the CCAN subunits could be the consequence rather than the cause of the mislocalization.
3. The authors hypothesize that a lack of SIMs is the underlying cause of the observation that sumoylation does not cause aggregation in this system, but disaggregation. Again, this is plausible, but the experiment addressing the SUMO interaction of the CCAN subunits are somewhat weak. A

lack of pull-down in this assay could be due to various reasons unrelated to SIM function, e.g. the accessibility of the proteins in the lysate etc).

4. Figure 4: Using RanGAP1 as a "control" substrate is not really legitimate because it is mainly a SUMO-1 substrate and is not known for poly-sumoylation.

Reviewer #2 (Remarks to the Author):

In this paper, Liebelt et al. describe the identification of poly-SUMOylated proteins by MS-based proteomics. The authors use their method based on pull down of His-tagged SUMO2 combined with knockdown of the poly-SUMO protease SENP6. Knockdown of SENP6 led to an accumulation of poly-SUMOylated proteins which were identified by quantitative proteomics leading to a list of 180 SUMO target proteins. Among these, many proteins of the constitutive centromere-associated network (CCAN) were present. Surprisingly, poly-SUMOylation of these proteins did not target them for ubiquitination and proteasomal degradation via RNF4, but instead prevent chromatin association and accumulation of CCAN proteins at centromeres.

The manuscript is well written and properly constructed, and the experimental data is of high quality, but I do have a few major concerns which I would like to see addressed.

Major comments:

1) Are all SENP6 substrates protected from degradation by RNF4 or is this a unique feature of the CCAN proteins? The authors do not show increased SUMOylation or ubiquitination for a selection of non-CCAN proteins that are present in their dataset (e.g. PML). Together with the fact that the authors fail to reproduce the RNF4-dependent degradation of CENP-I, this raises the question whether this has to do with the experimental toolbox and cellular system used? To make sure that quite severe genetic manipulation of the cells (expression of His10-SUMO2 + shRNA) do not lead to artificial cellular behavior, it seems essential that STUbL-mediated degradation of a few non-CCAN proteins can be confirmed (e.g. PML).

2) I agree that the data indicate that SENP6 and SENP7 have largely non-overlapping roles, however, I wonder which fraction of the total pool of poly-SUMOylated proteins is actually targeted by SENP6? Without adding proteomics data for SENP7 (I understand this is not within scope of the present

study), this could be addressed by showing SUMO2 western blotting data upon double KD of SENP6 and SENP7. Is the poly-SUMO smear still increasing after additional KD of SENP7 and how much? Related to this, could both proteases target different SUMO-linkages? Please comment on this in the discussion and briefly clarify for non-SUMO experts which linkages are known today (only K11?).

3) The proteomics strategy allows to measure differentially SUMOylated proteins. As with any PTM screen, it should be verified that parent protein levels are not changing upon SENP6 knock down and therefore the proteomics shotgun data of the input lysates should be added.

Other comments:

1. Page 6 and Figure 1: include western blotting data validating efficient SENP6 KD already in this figure (now only in Fig2B), ideally combined with SENP7 KD data (see above). Fig1B: indicate on blot where are di-SUMO moieties.

2. Include species information for all recombinant proteins, I assume most are human?

3. Page 6 and Figure 2: although published before, the SUMO2 purification strategy and the experimental setup should be briefly explained.

4. Page 7 and Figure 3: please explain how the GO analysis was performed.

5. Leave out legend for Figure 6C.

6. Page 10: tetramer should be trimer?

7. The experimental setup of the proteomics experiment is insufficiently described:

-Indicate how many replicates were analyzed for each condition, whether these were technical or biological replicates (different lentiviral infections?) and how many MS runs were finally performed. Also include this info in the legend of Fig 2A.

-Include a description of the LC-MS/MS analyses with most important parameters (MS instrument, run length, TopN settings, etc)

-Include a description of the MaxQuant search settings with most important parameters (search database, ppm settings,

-In Perseus “a series of two sample t-tests were performed”. Please describe which conditions were compared exactly and how this relates to the fold changes and p-values in Table S1 and Table S2 (include table legends).

-Please provide reviewer login info for the PRIDE dataset

8. Please extend the legends of all supplementary figures. Legends are minimal now and often not sufficient. A brief interpretation of the results should also be added. e.g. Figure 1SA: wat are EL1 and EL2?

Reviewer #3 (Remarks to the Author):

This manuscript examined regulation and the potential role of poly-SUMO chain by SENP6, a poly-SUMO chain targeting deSUMOylation enzyme. Authors utilized their established isolation and identification methods for SUMOylated proteins from cells by using His10 fused SUMO2 expressing cell lines. Combining shRNA-mediated knockdown of SENP6, they identified proteins which enriched their SUMOylated form in SENP6-dependent manner. The identification result showed a group of proteins that are involved in previously known SUMOylation-dependent cellular functions, including DNA repair, mitosis, and rRNA regulation. Then, they focused on CCAN proteins and discovered poly-SUMOylation on each target could have different consequences. Intriguingly, some of poly-SUMOylated CCAN proteins are not degraded by STUBL-mediated ubiquitination pathway, suggesting a potential novel function of poly-SUMO chain on substrate besides degradation. Investigation of cellular function SENP6-mediated regulation of poly-SUMO chain on CCAN proteins by SENP6 knockdown suggests that stabilized SUMO chain on these proteins resulted in defect on their association of chromosomes and failure to create functional centromeres for proper chromosome segregation. This finding provides conceptually novel role of SENP6-mediated

regulation of poly-SUMO chain on cellular proteins, as such it could control efficient formation of multi protein complex, for example CCAN complex in this case.

The role and regulation of SUMOylation of cellular proteins are a very important biological question because SUMOylation is an essential cellular PTM for all eukaryotes. Despite that importance, our understanding of regulation and function of SUMOylation is lacking as compared with those of ubiquitination. This reviewer appreciates the author's effort on the comprehensive analysis of SENP6 target proteins for poly-SUNOM chain regulation and their discovery of potential novel function of SUMO chain on CCAN regulation. My suggestions involving this manuscript further are mentioned below.

Major concerns.

1) To further support the finding with native condition, it would be preferable to examine the poly-SUMO chain formation (or SUMOylation) on their target proteins by using parental U2OS cells that does not have stable expression of His10 fused SUMO2. Although this might be challenging, it might be possible to isolate native SUMO2/3 modified proteins by using monoclonal SUMO2/3 antibody as previously reported under SENP6 knockdown.

2) To further confirm cellular function of SENP6 on CCAN, it would be ideal to perform a rescue experiment with shRNA-resistant SENP6 for CCAN defect observed in SENP6 knockdown. There are some inconsistencies between the two shRNAs which they used in figure 4&5. Therefore, the rescue might strengthen the biological relevance of the findings.

Minor point:

On page 5 line 15, Figure 1F should be Figure 3F.

Reviewers' comments:

Reviewer #1 (Remarks to the Author):

In this manuscript the authors report the results of a proteomics study designed to identify substrates of the SUMO-specific isopeptidase SENP6. Among their top hits they find subunits of the centromeric CCAN complex, in addition to factors involved in genome maintenance. Their data demonstrate that loss of SENP6 causes a delocalization of CCAN subunits to various different degrees, but that they are apparently not subject to RNF4-mediated ubiquitylation and targeting to the proteasome. From their results the authors conclude that the CCAN complex undergoes a process of group sumoylation/desumoylation and that group desumoylation of the complex by SENP6 is required for correct centromere localization. They thus postulate an alternative, non-degradative function of the poly-SUMO chains in preventing CCAN assembly.

The study is interesting in the sense that most polysumoylation events have been linked to RNF4-initiated degradation, whereas here the authors show that the CCAN subunits are apparently not targeted via this route. Otherwise, the study confirms previous data on the role of SENP6 in centromere assembly and mitosis, but it falls short of proving the authors' main claims: that it is the sumoylation of the CCAN subunits themselves that prevents them from engaging at the centromere. My specific points are as follows:

1. The Dasso lab has already shown that CCAN subunits don't correctly localize to the centromeres in SENP6-depleted conditions (Mukhopadhyay, 2006).

In reply: we have mentioned this in the text. Whereas the Dasso lab has shown that the CCAN subunit CENP-I is a SUMO target regulated by SENP6, our study indicates that nearly all CCAN subunits are SUMO-2 targets and affected by SENP6 knockdown, revealing striking functional protein group desumoylation.

2. The conclusion that the CCAN subunits are the relevant SUMO targets responsible for this phenotype is plausible, but the authors' experiments don't provide evidence for this. They exclusively work with SENP6 knockdowns and therefore cannot exclude the alternative model that sumoylation of a different SENP6 substrate is responsible for preventing centromere assembly of the CCAN complex. In fact, a publication that came out very recently in Cell Research (Fu et al. 2019: SENP6-mediated M18BP1 deSUMOylation regulates CENP-A centromeric localization) appears to argue along those lines. If that were the case, the hypersumoylation of the CCAN subunits could be the consequence rather than the cause of the mislocalization.

In reply: previously, the group of Mary Dasso claimed that CENP-A localization at the inner kinetochore was not affected by SENP6 knockdown "The amount of CENP-A on inner kinetochores was equivalent in the presence and absence of SENP6 (unpublished data)" (Mukhopadhyay et al. 2010 J. Cell Biol.). This initially discouraged us from investigating CENP-A localization upon SENP6 knockdown. After the submission of our manuscript, Fu et al. 2019 Cell Res. claimed that SENP6 is important for the centromeric localization of CENP-A via the regulation of Mis18BP1. We have also included in our first submission that Mis18BP1 is a SENP6 target and have shown this also in Figure 4. We have now confirmed that CENP-A centromeric localization is affected by SENP6 knockdown in the conditions used by us. This most likely occurs via deSUMOylation of Mis18BP1 by SENP6, since we found no evidence for

CENP-A sumoylation in the absence of SENP6 (Figure 4c). Therefore, we have revised our model by including Mis18BP1 and CENP-A as proteins affected directly and indirectly respectively by SENP6 knockdown.

Furthermore, it should be noted that the stabilization of CENP-A and CCAN at centromeres is a mutual phenomenon, since it has been shown previously that CCAN subunits are important for stabilization of CENP-A at centromeres (e.g. Hori et al. 2013 J. Cell Biol.; Falk et al. 2015 Science)

3. The authors hypothesize that a lack of SIMs is the underlying cause of the observation that sumoylation does not cause aggregation in this system, but disaggregation. Again, this is plausible, but the experiment addressing the SUMO interaction of the CCAN subunits are somewhat weak. A lack of pull-down in this assay could be due to various reasons unrelated to SIM function, e.g. the accessibility of the proteins in the lysate etc).

In reply: The buffer conditions used in our experiment include 150mM NaCl, physiological NaCl conditions. We had already included a positive control in this experiment, RNF4. To further strengthen our conclusions, we have now included another important positive control, SENP6 itself (Supplementary Figure 6b).

4. Figure 4: Using RanGAP1 as a "control" substrate is not really legitimate because it is mainly a SUMO-1 substrate and is not known for poly-sumoylation.

In reply: We have strengthened this part by adding topoisomerase II α and β as novel controls that are heavily sumoylated, but not increased for sumoylation upon SENP6 knockdown (Figure 4b).

Reviewer #2 (Remarks to the Author):

In this paper, Liebelt et al. describe the identification of poly-SUMOylated proteins by MS-based proteomics. The authors use their method based on pull down of His-tagged SUMO2 combined with knockdown of the poly-SUMO protease SENP6. Knockdown of SENP6 led to an accumulation of poly-SUMOylated proteins which were identified by quantitative proteomics leading to a list of 180 SUMO target proteins. Among these, many proteins of the constitutive centromere-associated network (CCAN) were present. Surprisingly, poly-SUMOylation of these proteins did not target them for ubiquitination and proteasomal degradation via RNF4, but instead prevent chromatin association and accumulation of CCAN proteins at centromeres.

The manuscript is well written and properly constructed, and the experimental data is of high quality, but I do have a few major concerns which I would like to see addressed.

Major comments:

1) Are all SENP6 substrates protected from degradation by RNF4 or is this a unique feature of the CCAN proteins? The authors do not show increased SUMOylation or ubiquitination for a selection of non-CCAN proteins that are present in their dataset (e.g. PML). Together with the fact that the authors fail to reproduce the RNF4-dependent degradation of CENP-I, this raises the question whether this has to do with the experimental toolbox and cellular system used? To make sure that quite severe genetic manipulation of the cells (expression of His10-SUMO2 + shRNA) do not lead to artificial cellular behavior, it seems essential that STUbL-mediated degradation of a few non-CCAN proteins can be confirmed (e.g. PML).

In reply: We found that the sumoylation of Mis18BP1 is further enhanced by double knockdown of SENP6 and RNF4 compared to SENP6 knockdown only. (Supplementary Figure 3a). We have also included PML as arsenic-induced RNF4 target as a positive control (Supplementary Figure 3b). This is working well, indicating that the lack of effect of RNF4 on CCAN proteins is not an artefact of our toolbox.

2) I agree that the data indicate that SENP6 and SENP7 have largely non-overlapping roles, however, I wonder which fraction of the total pool of poly-SUMOylated proteins is actually targeted by SENP6? Without adding proteomics data for SENP7 (I understand this is not within scope of the present study), this could be addressed by showing SUMO2 western blotting data upon double KD of SENP6 and SENP7. Is the poly-SUMO smear still increasing after additional KD of SENP7 and how much? Related to this, could both proteases target different SUMO-linkages? Please comment on this in the discussion and briefly clarify for non-SUMO experts which linkages are known today (only K11?).

In reply: We have compared single and double KDs of SENP6 and SENP7 and have verified high molecular weight SUMO2/3 signals by immunoblotting to verify high molecular weight smears. The data indicate that whereas SENP6 knockdown by itself does affect high molecular weight SUMO2/3, SENP7 knockdown does not (Figure 1c). The combination of SENP6 and SENP7 knockdown increases the high molecular weight SUMO2/3 signals, indicating that SENP6 can compensate for the absence of SENP7. We have added a speculative part to the Discussion, mentioning the idea that both proteases might target different SUMO-linkages. SUMO2 can in fact form chains via all internal lysines, with K11 as the most well-known linkage (Hendriks et al. 2017 Nat. Struct. Mol. Biol.)

3) The proteomics strategy allows to measure differentially SUMOylated proteins. As with any PTM screen, it should be verified that parent protein levels are not changing upon SENP6 knock down and therefore the proteomics shotgun data of the input lysates should be added.

In reply: We agree with the reviewer that it is important to verify parent protein levels to test if they are changing upon SENP6 knock down. However, in previous attempts, we have found that our proteomics set up (Q-Exactive) does not reach the full depth required to map full complexity cellular proteomes and immunoblotting is required for relevant target proteins. In light of this and since we focus on CCAN members and Mis18BP1 in our paper for follow-up experiments, we have verified input levels for these proteins. Inputs are shown in Figure 4 and don't show increases at the input level. Increases in SUMOylation for Mis18BP1 and CCAN members due to SENP6 knockdown can therefore not be explained by increases in total levels of these proteins.

Other comments:

1. Page 6 and Figure 1: include western blotting data validating efficient SENP6 KD already in this figure (now only in Fig2B), ideally combined with SENP7 KD data (see above). Fig1B: indicate on blot where are di-SUMO moieties.

In reply: we have added this in combination with SENP7 knockdown as requested for Figure 1c and indicated on the blot in Figure 1b the di-SUMO moieties

2. Include species information for all recombinant proteins, I assume most are human?

In reply: we confirm that all recombinant proteins are human.

3. Page 6 and Figure 2: although published before, the SUMO2 purification strategy and the experimental setup should be briefly explained.

In reply: we have added a brief explanation for the SUMO2 purification strategy.

4. Page 7 and Figure 3: please explain how the GO analysis was performed.

In reply: we have added a brief explanation for the GO analysis in the results section and a more detailed explanation in the methods section.

5. Leave out legend for Figure 6C.

In reply: we have left out the legend for old Figure 6c.

6. Page 10: tetramer should be trimer?

In reply: we thank the reviewer for pointing out this mistake; we have revised this.

7. The experimental setup of the proteomics experiment is insufficiently described:

-Indicate how many replicates were analyzed for each condition, whether these were technical or biological replicates (different lentiviral infections?) and how many MS runs were finally performed. Also include this info in the legend of Fig 2A.

In reply: we have added these details as requested.

-Include a description of the LC-MS/MS analyses with most important parameters (MS instrument, run length, TopN settings, etc)

In reply: we have added these details as requested.

-Include a description of the MaxQuant search settings with most important parameters (search database, ppm settings,

In reply: we have added these details as requested.

-In Perseus “a series of two sample t-tests were performed”. Please describe which conditions were compared exactly and how this relates to the fold changes and p-values in Table S1 and Table S2 (include table legends).

In reply: we have added these details as requested.

-Please provide reviewer login info for the PRIDE dataset

In reply: The mass spectrometry proteomics data have been deposited to the ProteomeXchange Consortium via the PRIDE partner repository with the dataset identifier PXD011963. The dataset can be accessed using a reviewers account with the username: reviewer06029@ebi.ac.uk and the password: Q4BtmjG.

8. Please extend the legends of all supplementary figures. Legends are minimal now and often not sufficient. A brief interpretation of the results should also be added. e.g. Figure 1SA: wat are EL1 and EL2?

In reply: we have extended the legends of all supplementary figures as requested.

Reviewer #3 (Remarks to the Author):

This manuscript examined regulation and the potential role of poly-SUMO chain by SENP6, a poly-SUMO chain targeting deSUMOylation enzyme. Authors utilized their established isolation and identification methods for SUMOylated proteins from cells by using His10 fused SUMO2 expressing cell lines. Combining shRNA-mediated knockdown of SENP6, they identified proteins which enriched their SUMOylated form in SENP6-dependent manner. The identification result showed a group of proteins that are involved in previously known SUMOylation-dependent cellular functions, including DNA repair, mitosis, and rRNA regulation. Then, they focused on CCAN proteins and discovered poly-SUMOylation on each target could have different consequences. Intriguingly, some of poly-SUMOylated CCAN proteins are not degraded by STUBL-mediated ubiquitination pathway, suggesting a potential novel function of poly-SUMO chain on substrate besides degradation. Investigation of cellular function SENP6-mediated regulation of poly-SUMO chain on CCAN proteins by SENP6 knockdown suggests that stabilized SUMO chain on these proteins resulted in defect on their association of chromosomes and failure to create functional centromeres for proper chromosome segregation. This finding provides conceptually novel role of SENP6-mediated regulation of poly-SUMO chain on cellular proteins, as such it could control efficient formation of multi protein complex, for example CCAN complex in this case. The role and regulation of SUMOylation of cellular proteins are a very important biological question because SUMOylation is an essential cellular PTM for all eukaryotes. Despite that importance, our understanding of regulation and function of SUMOylation is lacking as compared with those of ubiquitination. This reviewer appreciates the author's effort on the comprehensive analysis of SENP6 target proteins for poly-SUNOM chain regulation and their discovery of potential novel function of SUMO chain on CCAN regulation. My suggestions involving this manuscript further are mentioned below.

Major concerns.

1) To further support the finding with native condition, it would be preferable to examine the poly-SUMO chain formation (or SUMOylation) on their target proteins by using parental U2OS cells that does not have stable expression of His10 fused SUMO2. Although this might be challenging, it might be possible to isolate native SUMO2/3 modified proteins by using monoclonal SUMO2/3 antibody as previously reported under SENP6 knockdown.

In reply: Unfortunately, the immunoprecipitation efficiency of endogenous SUMO2/3 using the 8A2 monoclonal antibody was rather suboptimal in our hands. That was precisely the reason for us several years ago to develop alternative approaches. The yield and purity of any enrichment strategy are critical. We have carefully optimized the His10-SUMO2 methodology for this purpose. Nevertheless, we have carried out two attempts to immunoprecipitated endogenous SUMO2/3 using the 8A2 monoclonal antibody, with disappointingly low yield of about 1% at best, as shown below. The low yield of the SUMO2/3 immunoprecipitation unfortunately prevented the investigation of endogenous SUMO2/3 substrates.

To investigate whether additional evidence exists for the endogenous sumoylation of endogenous CCAN members and Mis18BP1, we have explored the literature and found evidence for many of them in Hendriks et al. 2018 Nature Communications, as shown in the table below. The evidence indicates that this functional protein group is indeed modified by endogenous SUMO2/3 and underlines that the sumoylation of centromeric proteins is not an artefact of our experimental system. However, even this approach could easily miss out on sumoylation sites that correspond to very large or very small tryptic peptides.

Uniprot	Protein name	Gene name	SUMO sites	Position in protein
Q6P0N0	Mis18-binding protein 1	Mis18BP1	30	1020;1034;1077; 1093;1108;126; 167;171;203;211; 262;379;534;54; 557;587;612;647; 65;676;688;694; 7;727;742;753; 850;899;96;99
Q03188	Centromere protein C (CENP-C)	CENPC	19	166;180;202; 205;212;217; 230;260;356; 399;45;507; 534;593;655; 677;698;807; 880
P07199	Major centromere autoantigen B (CENP-B)	CENPB	7	13;246;249; 266;276;58;584
Q9H3R5	Centromere protein H (CENP-H)	CENPH	2	67;81
Q7Z7K6	Centromere protein V (CENP-V)	CENPV	5	134;196;210; 217;270
Q02224	Centromere-associated protein E (CENPE)	CENPE	2	1731;373
Q9NQS7	Inner centromere protein	INCENP	3	256;327;454
Q9HC77	Centromere protein J (CENP-J)	CENPJ	1	790
Q5JX02	Centromere protein I (CENP-I)	CENPI	1	33
Q7L2Z9	Centromere protein Q (CENP-Q)	CENPQ	2	180;34
Q9BU64	Centromere protein O (CENP-O)	CENPO	4	102;14;290;33
Q71F23	Centromere protein U (CENP-U)	CENPU	2	244;63
Q8N0S6	Centromere protein L (CENP-L)	CENPL	1	29
Q5EE01	Centromere protein W (CENP-W)	CENPW	1	11
Q9NSP4	Centromere protein M (CENP-M)	CENPM	1	45

2) To further confirm cellular function of SENP6 on CCAN, it would be ideal to perform a rescue experiment with shRNA-resistant SENP6 for CCAN defect observed in SENP6 knockdown. There are some inconsistencies between the two shRNAs which they used in figure 4&5. Therefore, the rescue might strengthen the biological relevance of the findings.

In reply: *We agree with the reviewer that the rescue experiment is important to strengthen our findings. Indeed, the knockdown effect of SENP6 could be rescued by re-introducing shRNA-resistant wild-type SENP6, but not by shRNA-resistant catalytic-dead SENP6, - strengthening the biological relevance of our findings (Figure 8).*

Minor point:

On page 5 line 15, Figure 1F should be Figure 3F.

In reply: we thank the reviewer for pointing out this mistake and have corrected this.

REVIEWERS' COMMENTS:

Reviewer #1 (Remarks to the Author):

In their revised manuscript, the authors have included Mis18BP1 in their model of how group desumoylation by SENP6 affects recruitment of centromere proteins. This takes into account recent data published by Fu et al (2019). Yet, my main concern remains unsolved with this change: the lack of evidence what the functionally relevant SENP6 target(s) are. Since they exclusively work with SENP6 knockdown conditions, they have no means to differentiate whether desumoylation of the CCAN subunits themselves is actually important for their recruitment or relevant at all, or whether it is desumoylation of Mis18BP1 or of any other possibly unknown factor that allows their recruitment. As it stands, their data describe a phenomenon of group (de)sumoylation at the centromere and a function of SENP6 in this process. Whether the phenotype of SENP6 knockdown is related to this has not been shown, and whether the group sumoylation has any functional relevance remains unclear as well. Overall, the study expands on the data by the Dasso and Zhu labs and provides data that could indicate an interesting regulatory mechanism of protein recruitment if the model could be substantiated.

Most of my technical concerns have been answered, although even using 150 mM NaCl is not strictly speaking "physiological", as relative concentrations of the factors are not taken into account, and the "physiological" ion would be K rather than Na.

Reviewer #2 (Remarks to the Author):

The authors adequately addressed all my comments. I appreciate their efforts to include the necessary controls that were lacking in the first version of the manuscript. I have no further remarks.

Reviewer #3 (Remarks to the Author):

This revised manuscript from Frauke Liebelt et.al. satisfactory supports their conclusion that SENP6-mediated poly-SUMOylation can regulate CCAN proteins. In particular, rescue experiments of SENP6 siRNA knockdown by wild type SENP6 and catalytically-dead SENP6 mutants provide strong support

on their conclusion. It is unfortunate that technical difficulty to detecting endogenous SUMO2/3 modification by anti-SUMO2/3 antibody IP, I, however, think that other data satisfactory compensate that problem. I support accepting this revised version for publication.